

# Investigating Carbonyl Compounds above the Amazon Rainforest using PTR-ToF-MS with NO+ Chemical Ionization

Akima Ringsdorf[1], Achim Edtbauer[1], Bruna Holanda[2], Christopher Poehlker[2], Marta O. Sá[3], Alessandro Araújo[4], Jürgen Kesselmeier[2], Jos Lelieveld[1,5], Jonathan Williams[1,5]

[1]Department of Atmospheric Chemistry, Max Planck Institute for Chemistry, Mainz, Germany

[2]Department of Multiphase Chemistry, Max Planck Institute for Chemistry, Mainz, Germany

[3]Instituto Nacional de Pesquisas da Amazonia (INPA), Manaus, CEP 69067-375, Brazil

[4]Empresa Brasileira de Pesquisa Agropecuaria (Embrapa) Amazonia Oriental, Belem, CEP 66095-100, Brazil

[5]Climate and Atmosphere Research Center, The Cyprus Institute, 1645 Nicosia, Cyprus

*Correspondence to*: Jonathan Williams (J.Williams@mpic.de) and Akima Ringsdorf (A.Ringsdorf@mpic.de)

**Abstract**. The photochemistry of carbonyl compounds significantly influences tropospheric chemical composition by altering the local oxidative capacity, free radical abundance in the upper troposphere, and formation of ozone, PAN, and secondary organic aerosol particles. Carbonyl compounds can be emitted directly from the biosphere into the atmosphere and are formed through photochemical degradation of various precursor compounds. Aldehydes have atmospheric lifetimes of hours to days, in contrast to ketones, which persist for up to several weeks. While standard operating conditions for proton transfer time-of-flight mass spectrometer (PTR-ToF-MS) using $H_3O^+$ ions are unable to separate aldehydes and ketones, the use of $NO^+$ reagent ions allows for the differential detection of isomeric carbonyl compounds with a high time resolution. Here we study the temporal (24 h) and vertical (80–325 m) variability of individual carbonyl compounds in the Amazon rainforest atmosphere with respect to their rainforest-specific sources and sinks. We found strong sources of ketones within or just above the rainforest canopy (acetone, MEK, and $C_5$-ketones). A common feature of the carbonyls was nocturnal deposition observed by loss rates, most likely since oxidized volatile organic compounds are rapidly metabolized and utilized by the biosphere. With $NO^+$ chemical ionization, we show that the dominant carbonyl species include acetone and propanal, which are present at a ratio of 1:10 in the wet–to–dry transition and 1:20 in the dry season.

## 1 Introduction

On a global scale, tropical forests are regarded as the largest source of biogenic volatile organic compounds (BVOC) for the atmosphere (Guenther, 2013). BVOC comprise multiple compound classes including terpenes, alkenes, alkanes, alcohols, acids, esters, halocarbons, and carbonyls, all emitted as a result of various physiological processes, such as those occurring in plants, soils, etc., and as a function of environmental conditions. The emission quantity and composition vary among plant species, thus given the high biodiversity in tropical forests, the ecosystem composition and developmental stage also need to be considered as clearly demonstrated by Ciccioli et al. (2023). Most of the carbon released as BVOC from the tropical rainforest is in the form of terpenes, including the hemiterpene isoprene ($C_5H_8$) (Yáñez-Serrano et al., 2015), monoterpenes such as alpha-pinene ($C_{10}H_{16}$) (Zannoni et al., 2020b), and sesquiterpenes such as copaene ($C_{15}H_{24}$) (Yee et al., 2020). In addition, considerable amounts of oxygenated VOC (OVOC) are known to be present in rainforest air, with carbonyl compounds, namely aldehydes and ketones containing the C=O functional group, constituting an important subset of the atmospheric OVOC (Kesselmeier and Staudt, 1999). Direct biogenic emission, biomass burning, and secondary formation, mainly from the oxidation of the aforementioned terpene precursors and photolysis of larger carbonyls, all contribute to the cocktail of carbonyl compounds in the atmosphere (Guenther, 2013; Mellouki et al., 2015; Liu et al., 2022). To understand this cocktail, deposition and uptake by vegetation, i.e., bidirectional exchange, should always be considered as potential contributors (Kesselmeier, 2001; Kesselmeier et al., 1997; Villanueva et al., 2014). The formation of carbonyl species occurs after the oxidation of VOC is initiated by the hydroxyl radical (OH), ozone ($O_3$), or at nighttime by the nitrate radical ($NO_3$), and the resulting peroxy radicals ($RO_2$) react either with nitrogen oxide (NO) (when present) or with other ambient $RO_2$ or



HO$_2$ radicals. In the presence of NO this oxidation chain results in a net production of O$_3$, an important radiatively
active oxidant in the Amazon and worldwide (Mellouki et al., 2015; Trebs et al., 2012).
The main atmospheric carbonyl sinks are photolysis and oxidation by OH (Atkinson and Arey, 2003). As a
consequence, reactions with carbonyls combined with those of other BVOC determine the availability of OH and thus
the oxidative capacity of the atmosphere (Lelieveld et al., 2016). In Amazon rainforest air, OVOC account for 22-40 %
of OH reactivity, namely the overall loss frequency of OH radicals (Pfannerstill et al., 2021). Unsaturated carbonyls,
like the isoprene oxidation products methacrolein (MACR) and methyl vinyl ketone (MVK), are also oxidized by O$_3$.
Ketones, such as acetone, react much less readily with OH than aldehydes and, accordingly, have longer atmospheric
lifetimes. Thus, they persist during long-range transport and convective lifting to high altitudes, whereas more reactive
aldehydes impact the chemistry more locally. However, through rapid, deep convection, a frequent phenomenon in
the humid and hot tropics, also aldehydes can be transported to altitudes between 10 and 17 km (Prather and Jacob,
56  1997).

Oxidation of aldehydes and photolysis of ketones and dicarbonyls and further reaction with NO$_x$ (NO + NO$_2$) yields
peroxycarboxylic nitric anhydride (PAN). PAN and other peroxynitrates are thermally unstable near the surface but
in the cooler mid- and upper troposphere PAN is the most abundant reservoir for nitrogen oxides and is transported
over long distances (Mellouki et al., 2015; Fischer et al., 2014; Singh et al., 1990; Roberts, 2007). The main precursors
of PAN are acetaldehyde, followed by more minor contributions from methylglyoxal (not reported here) and acetone
(Fischer et al., 2014). NO$_x$ in the tropical atmosphere originates from several processes, starting with microbial
activities in soils and the release of NO, which is rapidly oxidized to NO$_2$, a large fraction even before it escapes the
canopy. NO$_2$ can be taken up by vegetation and only a part of this species traverses the canopy to the atmosphere
above (Breuninger et al., 2013; Rummel et al., 2002; Chaparro-Suarez et al., 2011). Further sources are lightning
discharges and biomass burning, the latter having the strongest seasonal variability (Bond et al., 2002).
The photochemical degradation of carbonyls in the atmosphere is also a source of HO$_x$ (HO$_2$ + OH) radicals,
particularly important in the upper troposphere where OH radical production from carbonyls can exceed primary
production in areas impacted by convection (Liu et al., 2022; Colomb et al., 2006; Lary and Shallcross, 2000; Prather
and Jacob, 1997). Furthermore, the abundance of radicals and oxidation products of carbonyls and dicarbonyls can
promote the formation and growth of secondary organic aerosols (Liu et al., 2022).
In this study, the observed diel and vertical (80-325 m) variability of 15 carbonyl species (C$_2$-C$_9$) was investigated.
These species were detected online with a PTR-ToF-MS using NO$^+$ as a reagent ion. This technique enables the
separation of isomeric aldehydes and ketones to identify their partitioning in the Amazonian atmospheric boundary
layer (ABL) at the ATTO site. Previous measurements of carbonyls have been conducted over the rainforest using
PTR-MS with H$_3$O$^+$ as the reagent ion(Yáñez-Serrano et al., 2016). With this method, both aldehyde and ketone
carbonyl forms are detected at the same mass. Usually, for airborne measurements, atmospheric chemists have argued
that the m/z used for the detection of C$_3$ carbonyls can be interpreted to be predominantly acetone since its atmospheric
lifetime is relatively long (Williams et al., 2001). However, near biogenic sources, the fractional distribution can be
different, and especially if the data is used to extract further information about the environment, such as OH
concentrations (Williams et al., 2000), the validity of this assumption should be verified.
The dataset presented here is the first online measurement of speciated individual aldehydes and ketones in the
Amazon. This rainforest environment is characterized by high solar insolation and vigorous vertical transport by deep
convection. In quantifying the relative abundance of carbonyl species, we aimed to improve the understanding of their
emissions, secondary formation in the atmosphere, transformation, and deposition in the Amazon rainforest region.
**2   Experimental**
**2.1   Measurement site and instrumentation**
All measurements were conducted at the Amazon Tall Tower Observatory (ATTO) within the primary tropical
rainforest of Brazil. The site is located 135 km NE of Manaus (02.14°S,58.99°W, 120 m above sea level) with the
main wind direction being NE to SE (Fig. S1). In the wet season (February–May), the air is typically nearly pristine



since the air masses pass over more than 1000 km of mostly unperturbed rainforest before being sampled, with a
possible influence due to long-range transport from African biomass burning pollution, which has been observed in
the beginning of the dry season (Fabruary–March) (Holanda et al., 2023). This is reflected by low concentrations of
$NO_x$ of less than 150 ppt in the ABL during the late wet season. In the dry season (August-November), however, air
influenced by mainly man-made biomass burning in South America was observed. In the same season enhanced black
carbon concentrations were measured due to the hemispheric wide summer maximum in biomass burning. The site
hosts a 325–m–tall tower and an 80–m walk-up tower, among other measurement and accommodation facilities. A
detailed map can be found in the Supplement (Fig. S2). The canopy height of the surrounding forest is about 35 m
(Kuhn et al., 2007). A comprehensive description of the site is provided by Andreae et al. (2015). The measurements
described here were conducted from June 23 until July 8 and from September 27 until October 14, 2019.
The sampling inlets for the BVOC measurements are located at 80, 150, and 325 m on the tall tower. Air is drawn by
high-volume pumps down to the instrumentation that is stored in an air-conditioned container at the foot of the tower.
By sequentially sampling each height for 5 minutes, a semi-continuous measurement can be achieved, so that each
height is sampled four times per hour. The flow in the insulated and heated (40 °C) Teflon sampling lines (3/8″OD)
is about $10\,l\,min^{-1}$. A long inlet line can be compared to a gas chromatographic column, which retains the sampled
VOC depending on their volatility and polarity, expressed by a wall saturation concentration (Pagonis et al., 2017).
Adsorption to the inner walls of the Teflon line caused a response time of 90 seconds at ATTO using a VOC gas
standard. Before the actual sampling of each height, the line was therefore flushed with ambient air to achieve
saturation. Tests with a 400–m inlet line in China have shown that the carbonyl compounds investigated in this study
have high saturation concentrations (C*, which is inversely proportional to the wall partitioning) and are not affected
by line loss (Li et al., 2023; Deming et al., 2019), but line effects such as a broadening of initially sharp concentration
peaks cannot be excluded. It has to be noted that sharp concentration peaks or spikes of short duration (< 90 s) were
not expected high above the homogenous vegetation of the rainforest. Some less volatile molecules, like
sesquiterpenes, never reached saturation and were additionally potentially degraded by $O_3$ or $NO_3$ (which was shown
to form inside the insulated tubing (Li et al., 2023)); thus, they were not detected. A potential contribution from the
oxidation of sesquiterpenes inside the tubing to detected carbonyl species cannot be excluded; however, this
contribution is expected to be minor given the rapidly decreasing sesquiterpene concentrations with increasing
distance from the canopy (Yee et al., 2018). The residence time in the tubing is short compared to the time that
sesquiterpenes are exposed to oxidation during atmospheric transport before reaching the sampling heights. VOC were
measured by a Proton Transfer Reaction Time of Flight Mass Spectrometer (PTR-ToF-MS 4000, Ionicon Analytik,
Innsbruck, Austria) (Jordan et al., 2009) with a time resolution of 20 seconds.
Meteorological data were measured at the walk-up tower at multiple heights up to 80 m (LI7500A, LI-COR
Biotechnology, Lincoln, USA) and at the tall tower at 325 m (Lufft, WS600-LMB, G. Lufft Mess- und Regeltechnik
GmbH, Fellbach, Germany) with a time resolution of 1 minute.

**2.2    NO+ chemical ionization**
PTR-ToF-MS in general is a form of chemical ion mass spectrometry (CIMS) commonly operated with hydronium
ions ($H_3O^+$) for the chemical ionization of VOC in air samples. The technique is well-established and sensitive and is
able to detect most of the prominent VOC in ambient air with a high temporal resolution of seconds (de Gouw and
Warneke, 2007). The proton transfer reaction that lends its name to the instrument occurs between $H_3O^+$ ions and the
molecules R with a higher proton affinity than water (> 691 kJ $mol^{-1}$) (Hunter and Lias, 1998).

132           $$R + H_3O^+ \longrightarrow RH^+ + H_2O \qquad\qquad (R1)$$

Thus, isomeric molecules (such as acetone and propanal) form the same product ion $RH^+$ and cannot be distinguished.
For the purpose of investigating the atmospheric chemistry of carbonyl compounds, this is a major disadvantage since
the distribution between short-lived aldehydes and longer-lived ketones with the same carbon number remains unclear.
However, it has been shown that by using an alternative reagent ion (i.e., $NO^+$), aldehydes and ketones can be



distinguished. $NO^+$ ionizes aldehydes mainly via hydride abstraction (R2), whereas ketones and $NO^+$ tend to form a
cluster (R3) leading to different product ions (Koss et al., 2016; Španěl et al., 1997).

139            $R + NO^+ \longrightarrow (R\text{-}H)^+ + HNO$          (R2)

140            $R + NO^+ \longrightarrow (RNO)^+$             (R3)

To implement the $NO^+$ chemical ionization mass spectrometer ($NO^+$ CIMS), synthetic air instead of water vapor was
introduced in the ion source, and the source parameters were tuned to achieve a low contribution of impurity ions
($H_3O^+$, $O_2^+$, $NO_2^+$) and high counts of $NO^+$. The identity of the reaction that occurs to ionize the target compound
depends on the thermodynamical properties of the molecule. The hydride ion affinity of aldehydes is less than that of
$NO^+$, so R2 is exothermic and favored (Karl et al., 2012; Španěl et al., 1997). Ketones do not show the same tendency
to donate a hydrogen atom and the ionization energies of most ketones, especially small ones, is slightly higher than
that of NO ($> 9.26$ eV) (Smith et al., 2003). Thus, an association reaction, R3, primarily occurs for the ketones in this
study. Due to the humid conditions in the rainforest, $NO^+(H_2O)$-clusters were also available to react with ketones via
ligand switching, producing the same products as the association reaction R3 (Smith et al., 2003). The ionization
energies of 3-hexanone, 2-heptanone, and 2-nonanone are smaller than or equal to that of NO; nevertheless, the
association reaction has been shown to be favored by selected ion flow tube (SIFT) studies (Španěl et al., 1997). Those
compounds were, however, not detected in the mass spectra obtained in the rainforest environment examined in this
study.
Besides the most favored reaction, other ionization channels can also produce product ions. This, and partial
fragmentation in the drift tube can lead to additional complications of the mass spectra. To identify the distribution of
product ions and fragments of carbonyls for the type of instrument used in this study, a single-compound headspace
analysis was performed in the laboratory using a PTR-ToF-MS 8000. The basic components of the instrument, mainly
the ion source, drift tube, and detector are similar to a PTR-ToF-MS 4000, so that the relative transmission can be
assumed to be identical. The instrument was tuned to have the same E/N (electric field intensity divided by gas number
density) in the drift tube and similar impurities ($\leq 5$ %) as the instrument in the field. In both field and laboratory, two
different settings for the E/N values were applied. One set had a relatively low E/N of 70 Td, which has been
recommended in previous studies to minimize fragmentation (Koss et al., 2016; Romano and Hanna, 2018); the other
was operated with 120 Td for comparison. The results of the single-compound headspace analysis can be found in the
Supplementary (Table S1).
The complexity of the mass spectra measured with a $NO^+$ CIMS is a disadvantage if one aims for a non-targeted
analysis of VOC present in a certain environment, such as the rainforest. Long-term VOC observations at ATTO are
therefore conducted with a PTR-ToF-MS using $H_3O^+$ ions. However, for a targeted analysis, specifically for separating
carbonyl compounds, the $NO^+$ CIMS is a convenient method (Koss et al., 2016; Karl et al., 2012; Ernle et al., 2023).
Another advantage of the $NO^+$ chemistry is the ability to detect certain alkanes, as their proton affinity is too low to
be detected by a PTR-MS (Koss et al., 2016). This has been widely used in urban or rural areas to quantify vehicle
emissions, but such species have not yet been investigated at the ATTO rainforest site (Wang et al., 2020a; Chen et
al., 2022).

### 2.3    VOC data analysis

Integration of the mass spectra, baseline-, and duty-cycle-correction were performed using the IDA software (Ionicon
Analytik). In a subsequent step, the obtained signals were normalized to $NO^+$ and $H_2ONO^+$ and drift parameters like
pressure and temperature, to account for fluctuations.
Table 1 shows the sensitivities and limits of detection (LoD) for all target molecules with E/N values of 70 and 120 Td
applied. It was evident that the sensitivity of ketones decreases dramatically with high E/N conditions, most probably
due to enhanced fragmentation caused by more collisions in the drift tube.



Compounds displayed in bold in Table 1 were quantified using a primary VOC gas standard (Apel-Riemer
Environmental Inc., Colorado, USA). Unfortunately, this did not comprise all target carbonyls, and for those
compounds not in the standard, a theoretical method was applied to obtain concentrations, resulting in a higher
uncertainty. The relative distribution of the product ions obtained from the single-compound headspace analysis was
used to correct for the fragmentation of carbonyl compounds with higher m/z-ratios onto the parent m/z-ratios of other
target compounds.
For those compounds not included in the gas standard, mixing ratios were obtained by calculating the ionization
efficiency with a previously determined reaction rate of NO$^+$ and the target compound under the current conditions in
the drift tube (k-rate analysis) (Cappellin et al., 2012). The reaction rates (k-rates), also presented in Table 1, have
been derived for the sum of all product ions. Thus, a weighting factor for the relative production of the target ion
needed to be applied, which was also obtained by the single-compound headspace analysis from the slope of the
signals of the target ion vs. other product ions. The mixing ratios of both E/N settings, obtained by applying the
respective product ion distributions, agree well for most compounds (except for n-hexanal and ketones, which have a
low sensitivity at 120 Td). This accordance supports the assumption that product ion distributions were valid for both
instruments. To calculate propanal, the calibration factor of methacrolein was used, since in a previous calibration
measurement with the PTR-ToF-MS 8000 both compounds had similar sensitivities (methacrolein: 0.13 ppb ncps$^{-1}$,
propanal: 0.17 ppb ncps$^{-1}$).
The measurement uncertainty in the mixing ratios of standard calibrated VOC was less than 25%. It was derived from
the accuracy of the VOC gas standard ($\pm$ 5%), the flow meter used for the calibration ($\pm$ 1%), the accuracy of the least
square fit of the calibration curve (molecule-dependent, circa $\pm$ 10%), and the uncertainty of the relative distribution
of product ions, which was expected to be below 20%. The uncertainty of the product ion distribution was estimated
from the purity of the liquid carbonyls tested ($>$ 95%) as well as possible contamination during the headspace
sampling. In the case of theoretically calculated mixing ratios using k-rates the accuracy was accordingly higher. The
accuracy of the k-rate ($\pm$ 20%) (Španěl et al., 1997) and the accuracy of the distribution of product ions give the
absolute accuracy for k-rate calibrated mixing ratios which was thus estimated to be below 30%.
Detection limits were defined as three times the standard deviation of the background noise at the specified mass.
Those are also displayed in Figures 1–2. Negative values arising from the subtraction of the background were set to
zero to account for a slightly too high background measurement of some compounds during calibration.

## 2.4    Validation of observations

Pre-separation of the VOC with a GC column prior to detection with the NO$^+$ CIMS can indicate the pureness or
compound specificity of an m/z ratio. Koss et al., 2016 reported such data for urban ambient air and concluded that
certain masses can be seen as unambiguous in that environment. The E/N field used in that study, which strongly
impacts the fragmentation patterns on different m/z ratios, was similar to this study (60 Td), but the measurement site
was a parking lot in an urban area (Koss et al., 2016). Uncontaminated m/z ratios assigned to carbonyl compounds
were found for acetaldehyde, propanal, methacrolein, and crotonaldehyde, the sum of C$_5$-aldehydes, acetone, hexanal,
MVK, methyl ethyl ketone (MEK), benzaldehyde, heptanal, the sum of C$_5$-ketones, and octanal. Nevertheless,
biogenic compounds that may not be present in an urban environment were, therefore, not part of the GC method
applied in Koss et al. and remained as potential interferents for the carbonyl m/z ratios.
Allyl ethyl ether, an isomer of C$_5$-carbonyls that also undergoes hydride transfer, was potentially such a candidate for
interfering in the C$_5$-aldehyde m/z ratio. (Smith et al., 2011; Španěl and Smith, 1998). The m/z ratio of C$_5$-aldehydes
might have also been affected by 1-5 pentanediol if present at significant concentrations (Španěl et al., 2002). Some
carboxylic acids react with NO$^+$ under the drift tube conditions to form R $-$ OH $+$ HNO$_2$ and thus make isomers to the
ionized carbonyl species. Trimethylacetic acid was reported to mainly form C$_5$H$_9$O$^+$ and thus can also potentially
interfere with C$_5$-aldehydes (Španěl and Smith, 1998).
N-butyric acid is part of the glucose metabolism in plants and, upon ionization, partly makes C$_4$H$_7$O$^+$ ions (m/z
71.0491); thus, it potentially interfered with butanal (Smith et al., 2011). The same holds for isobutyric acid. Also,



valeric acid has been shown to fragment into $C_4H_7O^+$ to a great extent (Ŝpaněl and Smith, 1998). For the alcohols 2-
butanol, 1,4-butanediol, and the ester methyl butyrate, fragmentation into $C_4H_7O^+$ has been shown to occur (Koss et
al., 2016; Španěl et al., 2002; Ŝpaněl and Smith, 1998). Tetrahydrofuran, an ether isomeric with butanal is ionized via
hydride transfer and also forms $C_4H_7O^+$ (Španěl and Smith, 1998). Contamination from 2-butanol was shown to
account for around 50% of $C_4H_7O^+$ at an urban site in Boulder, USA (Koss et al., 2016). Since 2-butanol has been
previously found in emissions from vegetation (Kesselmeier and Staudt, 1999) and the mixing ratios of $C_4H_7O^+$ were
close to the detection limit, butanal could not be investigated without potential bias from other oxygenated VOC. With
another measurement technique (sampling to adsorbent tubes and measurement with a GC-ToF-MS) applied at ATTO
also no significant butanal peak was found. However, butanal has been identified in the Amazonian atmosphere during
the dry and wet seasons at another site in 1999 (Andreae et al., 2002).



**Table 1: List of identified carbonyl compounds and other hydrocarbons and their properties for detection with NO⁺ CIMS (PTR-ToF-MS 4000). Sensitivities are compared to the classical PTR-MS method using H₃O⁺ reagent ions. The "product factor" represents the weighting factor for the k-rate obtained from the distribution of product ions as described in section 2.3. Compounds in bold were quantified using a primary standard.**

| | | | | NO⁺ | | | | | | H₃O⁺ |
|---|---|---|---|---|---|---|---|---|---|---|
| | | | | E/N = 70 Td | | | E/N = 120 Td | | | E/N = 120 Td |
| Carbonyl species | Ion formula | Exact m/z | k-rate $10^{-9}$ cm³ s⁻¹ | Prod. factor | Sensitivity ppb ncps⁻¹ | LoD ppb | Prod. factor | Sensitivity ppb ncps⁻¹ | LoD ppb | Sensitivity ppb ncps⁻¹ |
| **Acetaldehyde** | $C_2H_3O^+$ | 43.01784 | 0.6(Španěl et al., 1997) | - | 0.155 | 0.112 | - | 0.431 | 0.160 | 0.025 |
| **Acetone** | $C_3H_6NO_2^+$ | 88.0393 | 1.2(Španěl et al., 1997) | (0.43) | 0.078 | 0.06 | 0.27 | 4.803 | 0.705 | 0.031 |
| **Propanal** | $C_3H_5O^+$ | 57.0335 | 2.5(Španěl et al., 1997) | (0.79) | *0.046* | 0.053 | 0.82 | *0.256* | 0.049 | - |
| **MEK** | $C_4H_8NO_2^+$ | 102.055 | 2.8(Španěl et al., 1997) | (0.84) | 0.049 | 0.008 | 0.61 | 1.027 | 0.111 | 0.028 |
| MVK | $C_4H_6NO_2^+$ | 100.039 | 2.4(Michel et al., 2005) | 0.86 | - | 0.004 | 0.67 | - | 0.012 | - |
| **MACR** | $C_4H_5O^+$ | 69.03349 | 2.6(Michel et al., 2005) | (0.54) | 0.046 | 0.021 | 0.42 | 0.256 | 0.093 | 0.028 |
| n-pentanone | $C_5H_{10}NO_2^+$ | 116.0706 | 3.4(Španěl et al., 1997) | 0.85 | - | 0.005 | 0.56 | - | 0.007 | - |
| n-pentanal | $C_5H_9O^+$ | 85.0648 | 3.2(Španěl et al., 1997) | 0.79 | - | 0.003 | 0.28 | - | 0.011 | |
| n-hexanone | $C_6H_{12}NO_2^+$ | 130.0863 | 3.3(Španěl et al., 1997) | - | - | 0.002 | - | - | - | - |
| Hexanal | $C_6H_{11}O^+$ | 99.0804 | 2.5(Španěl et al., 1997) | 0.75 | - | 0.006 | 0.4 | - | 0.016 | - |
| Trans-2-hexenal | $C_6H_9O^+$ | 97.0672 | 2.8(Roberts et al., 2022) | 0.68 | - | 0.006 | 0.82 | - | 0.005 | - |
| Benzaldehyde | $C_7H_5O^+$ | 105.033 | 2.8(Španěl et al., 1997) | 0.96 | - | 0.005 | 0.97 | - | 0.003 | - |
| Heptanal | $C_7H_{13}O^+$ | 113.0961 | 2 | - | - | 0.004 | - | - | 0.007 | - |
| Octanal | $C_8H_{15}O^+$ | 127.1117 | 2.7(Romano and Hanna, 2018) | 0.81 | - | 0.004 | 0.61 | - | 0.004 | - |
| Nonanal | $C_9H_{17}O^+$ | 141.1274 | 1.1(Roberts et al., 2022) | 0.04 | - | 0.145 | 0.1 | - | 0.078 | - |
| Nopinone | $C_9H_{14}O^+$ | 138.1039 | 2 | - | - | 0.019 | - | - | 0.002 | - |
| *Alkanes* | | | | | | | | | | |
| Isopentane | $C_5H_{11}^+$ | 71.086 | 2 | - | - | 0.013 | - | - | 0.027 | - |
| Methyl-cyclopentane | $C_6H_{11}^+$ | 83.086 | 2 | - | - | 0.005 | - | - | 0.008 | - |
| 2-, 3-methyl-pentane | $C_6H_{13}^+$ | 85.101 | 2 | - | - | 0.007 | - | - | 0.006 | - |
| $C_7$ cyclic alkanes | $C_7H_{13}^+$ | 97.101 | 2 | - | - | 0.004 | - | - | 0.003 | - |



| VOC species | Ion formula | Exact m/z | k-rate $10^{-9}$ cm³ s⁻¹ | NO⁺ | | | | | | H₃O⁺ |
| | | | | E/N = 70 Td | | | E/N = 120 Td | | | E/N = 120 Td |
| | | | | Prod. factor | Sensitivity ppb ncps⁻¹ | LoD ppb | Prod. factor | Sensitivity ppb ncps⁻¹ | LoD ppb | Sensitivity ppb ncps⁻¹ |
|---|---|---|---|---|---|---|---|---|---|---|
| C2-alkyl-cyclohexanes | $C_8H_{15}^+$ | 111.117 | 2 | - | - | 0.004 | - | - | 0.005 | - |
| *Alkenes* | | | | | | | | | | |
| C₅-alkene (2-pentenes) | $C_5H_{10}^+$ | 70.0777 | 2 | - | - | - | - | - | 0.009 | - |
| C₅-alkene (a-olefin) | $C_5H_{10}NO^+$ | 100.076 | 2 | - | - | 0.006 | - | - | 0.003 | - |
| $C_6H_{10}$ | $C_6H_{10}^+$ | 82.0777 | 2 | - | - | 0.006 | - | - | 0.01 | |
| *Alcohols* | | | | | | | | | | |
| Ethanol | $C_2H_5O^+$ | 45.0335 | 2 | - | - | 0.050 | - | - | 0.019 | - |
| *Alkyne* | | | | | | | | | | |
| Propyne | $C_4H_6^+$ | 54.046 | 2 | - | - | 0.026 | - | - | 0.011 | - |
| *Aromatic* | | | | | | | | | | |
| Benzene | $C_6H_6^+$ | 78.046 | - | - | 0.101 | 0.020 | - | 0.071 | 0.009 | 0.063 |
| *Terpenes* | | | | | | | | | | |
| Isoprene | $C_5H_8^+$ | 68.0621 | - | - | 0.078 | 0.018 | - | 0.068 | 0.023 | 0.045 |
| Sum of mono-terpenes | $C_{10}H_{16}^+$ | 136.125 | - | - | 0.067 | 0.004 | - | 0.554 | 0.039 | 0.103 |
| *Other* | | | | | | | | | | |
| Furan | $C_4H_4O^+$ | 68.0258 | 2 | - | - | 0.008 | - | - | - | - |
| $C_5H_4O_3$ | $C_5H_4NO_4^+$ | 142.014 | 2 | - | - | 0.005 | - | - | 0.003 | - |

Propionic acid is a potential contaminant for propanal on the m/z of $C_3H_5O^+$, but only a fraction of the acid was found
to land on the propanal m/z (Ŝpaněl and Smith, 1998). A higher fraction of the fragments of methyl and ethyl
propionate were detected as isomers to ionized propanal but have not been found to be present in biogenic emissions
so far (Ŝpaněl and Smith, 1998; Kesselmeier and Staudt, 1999).
It can also not be excluded that fragmentation to $C_2H_3O^+$ of several species, in particular acetic acid, methyl formate,
methyl acetate, and ethyl acetate contributes to the m/z ratio of acetaldehyde ($C_2H_3O^+$). Experimental evidence for the
contamination has only been found for a small contribution of methyl and ethyl acetate of less than 20% (Ŝpaněl and
Smith, 1998).
The isomers hexanal and z-3-hexenol are known to be emitted together by damaged green leaves (Langford et al.,
2010; Jardine et al., 2012a). A possible detection of both compounds on m/z of $C_6H_{11}O^+$ could not be excluded, since
alcohols also undergo hydride abstraction during the reaction with NO⁺ (Koss et al., 2016).
To our knowledge, none of the species that were demonstrated to fragment on the same m/z ratios as carbonyls have
been reported to be abundant in forested environments or even to be biogenically emitted, except for z-3-hexenol, 2-
butanol, n- and isobutyric acid, acetic acid, and propionic acid. In general, acids have primary sources, including
biogenic emissions and biomass burning but also photochemical sources including the ozonolysis of alkenes
(Orzechowska et al., 2005). The dataset from this study and comparison with the corresponding m/z of acids under





$H_3O^+$ ionization that have been measured previously at the ATTO site suggested that carboxylic acids undergo an
association reaction with $NO^+$. A headspace analysis with acetic acid also revealed no significant contributions to any
other m/z except the association product $C_2H_4NO_3^+$.
Fragmentation from higher carbonyls to m/z ratios attributed to lower carbonyls was observed in the single compound
headspace analysis, conducted with aldehydes and ketones up to nonanal. The m/z of acetaldehyde ($C_2H_3O^+$, 43.0178)
saw small contributions from acetone and pentanone, which were subtracted from the acetaldehyde signal. For this
correction, the relative contribution of the fragments from their parent mass, which was determined by the headspace
analysis, was used. A list of the single compounds and their product ions formed in the drift tube can be found in the
supplementary Table S1. Contributions from higher carbonyls in the $NO^+$ CIMS were not likely since they were not
observed or were below the detection limit.

## 3   Results

### 3.1   Atmospheric conditions and seasonality

Seasonality in the central Amazon is characterized by a comparatively less polluted wet season (February–May) and
a more strongly polluted dry season, due to the more frequent influence of biomass burning (August–November)
(Pöhlker et al., 2019; Holanda et al., 2023). The $NO^+$ CIMS measurements took place from June 23 until July 8 and
from September 27 until October 14, 2019. Below, we outline the meteorological conditions during both measurement
periods as they influenced seasonal variations in observed VOC mixing ratios and correlations. It is important to
consider that the photochemical loss of VOC and reactions involving OH depend on the availability of sunlight, which
also affects the secondary formation of OVOC from the oxidation of different hydrocarbons. VOC emissions from
vegetation are driven by light (photosynthetically active radiation, PAR), temperature, water availability, air pollution,
and biotic factors, such as herbivore infestation, pathogenic infections, or the developmental stage of a plant
(Laothawornkitkul et al., 2009). However, at heights above 80 m, integrated VOC emissions from a whole forested
area domiciled by various plant and herbivorous species at all developmental stages were sampled. As has been
reported previously, inter-seasonal growth variations may even induce the plant to switch from isoprene emission to
monoterpene emission and back (Kuhn et al., 2004a, b). The growth of new leaves (leaf flush), which are
photosynthetically more effective than mature leaves peaks in the dry season and is correlated with the availability of
light (Restrepo-Coupe et al., 2013), which causes an inter-seasonal gradient possibly manifested in the presented
BVOC emissions. The emission and uptake of BVOC by soils and cryptogamic organisms was shown to depend on
the availability of water and could additionally contribute to observed seasonal differences in BVOC concentrations
(Bourtsoukidis et al., 2018; Edtbauer et al., 2021).
On average, daytime temperatures differed by only 0.4 °C between the transition (June – July) and the dry season
(Fig. S3). Maximum temperatures in the canopy (at 26 m) were reached at 12:00 local time (LT), with 30.5 °C in the
transition season and 31.2 °C in the dry season on average. The diurnal evolution of temperature closely followed the
incoming solar radiation, here represented by PAR. Dry season observations of PAR were higher by about 9%
compared to the transition season. Precipitation in the month before the $NO^+$ CIMS measurements took place totaled
157 mm in June and 119 in September 2019 (Fig. S4). The water level measured in the Rio Negro close to Manaus in
2019, however, exhibited maximum values in June and minimum values in October, with a difference of about 10 m
(Chevuturi et al., 2022).
The sampled air originated predominantly from the east (SE to NE); thus, an influence from the city of Manaus could
be excluded (Fig. S1). However, for long-lived anthropogenic alkanes, influence from populated areas along the
Amazonas and smaller side rivers was conceivable. The detected alkanes (Table 1) had low mixing ratios below the
detection limit, indicating no significant influence from industries based on fossil fuel combustion.
Black carbon (BC) was used as a marker of biomass burning emissions. BC sampled at ATTO has been shown to
originate from biomass burning in South America and Africa (Holanda et al., 2020, 2023). Enhanced concentrations
of 0.42 and 0.54 µg m$^{-3}$ (80, 325 m) were found on average in the 2019 dry season. Maximum concentrations reached



0.93 and 1.17 µg m$^{-3}$. Average concentrations of 0.18 and 0.21 BC (80, 325 m) in the transition season indicated less
polluted conditions. A large number of VOC, including certain carbonyl compounds, are usually co-emitted during
biomass burning with various emission factors and rates (Andreae and Merlet, 2001; Andreae, 2019). Therefore, the
carbonyls detected with the NO$^+$ CIMS during this study and their precursors potentially originated from both biogenic
and biomass burning sources. Correlations of carbonyls with BC at 325 m are shown in the Supplementary Data for
both seasons (Fig. S5) to detect possible influences from advected, fresh, or aged biomass burning plumes. In the cases
of acetaldehyde, acetone, methacrolein, MVK, and benzaldehyde, a Pearson coefficient of $p > 0.55$ was calculated for
the day and nighttime so that an influence of biomass burning through co-advection or in plume production was
feasible.

### 3.2    Vertical distribution of carbonyls above the canopy

The distribution of carbonyls with height above the uniform rainforest-covered landscape provides information on the
nature of emission sources, oxidative transformations, and carbonyl sinks under consideration of dynamic processes
in the atmospheric mixed layer. Vertical gradients were governed by the strength and temporal variance of the
respective source, the atmospheric lifetime of the species considered, and dilution through turbulent mixing or
entrainment from the free troposphere during mixed layer growth. Earlier work investigated the chemical and dilutive
loss of isoprene with height using observations at ATTO and a turbulence-resolving large eddy simulation (DALES).
It was shown that slightly more than 50% of the isoprene loss in the vertical (80–325 m) at noon occurred due to
dilutive turbulent mixing (Ringsdorf et al., 2023). It is important to note that the lowest sampling height at 80 m was
within the roughness sublayer. This is a layer within the mixed ABL of about 2–3 times the canopy height (≈ 35 m),
which is strongly affected by the tall canopy with respect to wind fields and, thus, turbulence. Within this layer, the
exchange between the canopy and atmosphere occurs by inhomogeneous flows into and out of the canopy (Chamecki
et al., 2020). An important process influencing the ambient concentration of the compounds presented at all sampling
heights was the growth of the ABL (up to 2 km height) after sunrise due to the strengthening of turbulences from
thermal expansion of the heated air masses near the ground. During ABL growth, air from higher altitudes (residual

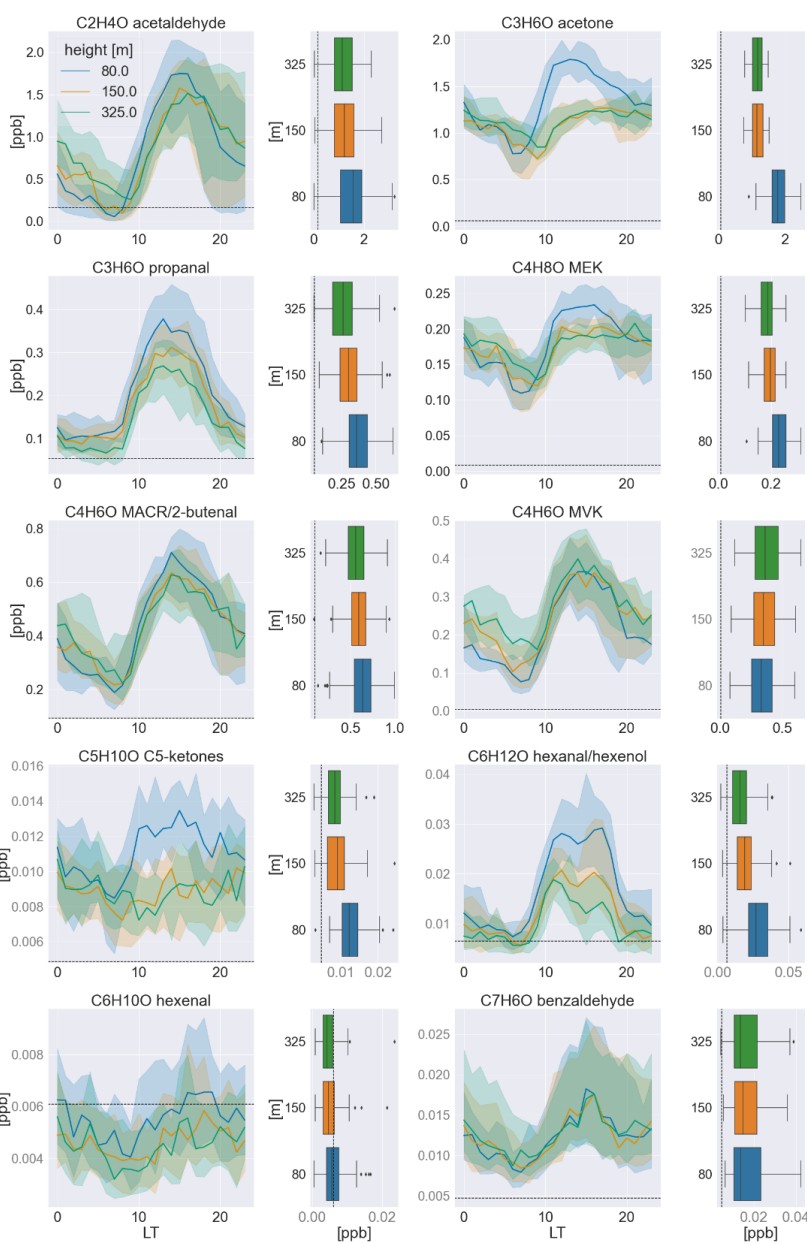


**Figure 1: Median averaged timeseries in the wet–to–dry transition season (June/July) of 2019 measured at all sampling heights for each carbonyl compound and its respective vertical profile at noon (12:00–15:00 LT) to the right. The shadings indicate the quartiles (25th and 75th). In the box-and-whisker plots, the boxes also represent the quartiles, while the residual data except for outliers are included in the whiskers. The detection limit (3 sigma) is indicated by dashed, black lines. The mixing ratios in black font were calibrated to a standard, while those in gray font were calculated based on the k-rate.**






layer containing more chemically aged air) is entrained, leading to the minimum mixing ratios observed after 06:00 LT
at all three heights. During the day, turbulent mixing via convection and associated downward motions is strongest
until convection eases with decreasing insolation. At night, a stable stratification associated with low vertical mixing
is formed (Jordi Vilà-Guerau de Arellano, et al., 2015).
Under the reasonable assumption of a carbonyl source at canopy level (based on emission inventories discussed in
Chapter 4), the long-lived ketones were expected to have a background concentration in the convective mixing layer
but also above, while levels of short-lived aldehydes will tend to be zero at higher altitudes, like for isoprene.
Consequently, the aldehydes should show a stronger decrease in their vertical profiles than the ketones, which were
expected to be well-mixed at about a hundred meters above the canopy throughout the convective mixing layer.
Nonetheless, one has to also take the secondary chemical formation of carbonyls into account, which can influence
the vertical gradients depending on the emission source and atmospheric lifetime of the precursors.
Figures 1 and 2 present the diurnal cycle observed at the three sampling heights for all carbonyls measured in the wet–
to–dry transition and the dry season, respectively. Some compounds were measured in very low concentrations, below
the detection limit in one or both seasons, namely, the sum of $C_5$-aldehydes, $C_6$-ketones, heptanal, octanal, and
nonanal. All other carbonyls showed distinct diurnal variabilities with increasing concentrations after sunrise
(06:00 LT) and decreasing concentrations at nighttime. Their diurnal cycle followed the evolution of PAR and
temperature with a slight delay throughout the day, reflecting the expected biogenic emission and photochemical
production. As hypothesized above, no significant vertical variability was found for ketones, though only at 150 and
325 m, whereas a strong decrease in mixing ratio with height was observed between 80 and 150 m. This distribution
indicates that mixing ratios of ketones were only well-mixed above 150 m, while the measurements at 80 m were
influenced by a strong source of ketones, which is discussed compound-wise below. The observed aldehydes exhibited
different vertical distributions; some showed increasing mixing ratios with height, others were rather steadily
decreasing as it was hypothesized, and some showed very small variabilities throughout the lowermost 325 m of the
atmosphere.

**3.3    Correlations at 80m and common sources**
The chemical composition of airmasses measured at 80 m was governed by various processes occurring from the leaf
level up to mixing scales of the lower atmosphere. At the leaf level, BVOC are formed by plant metabolic pathways
or, possibly, in the case of OVOC, including carbonyls via within-leaf reactions. Epicuticular waxes, also called leaf
waxes, consist of long-chained hydrocarbons, e.g., the triterpene squalene, which yield OVOC during ozonolysis.
Depending on the position of the double bond of the long-chained molecule and its functional groups, aldehydes or
ketones are formed, whereby the chances for the formation of short-chained carbonyls like acetone are highest
(Fruekilde et al., 1998). Following their emission, a fraction is deposited on surfaces, which is in most cases reversible,
or taken up by stomata, which represents a potential sink (Niinemets et al., 2014). Depending on their atmospheric
lifetime, BVOC undergo within-canopy oxidation; in the case of reactive isoprene and monoterpenes, this was found
to come to not more than 10% of their initial emission flux (Karl et al., 2004; Fuentes et al., 2022) before being ejected
from the canopy. Within and above the canopy they are mixed



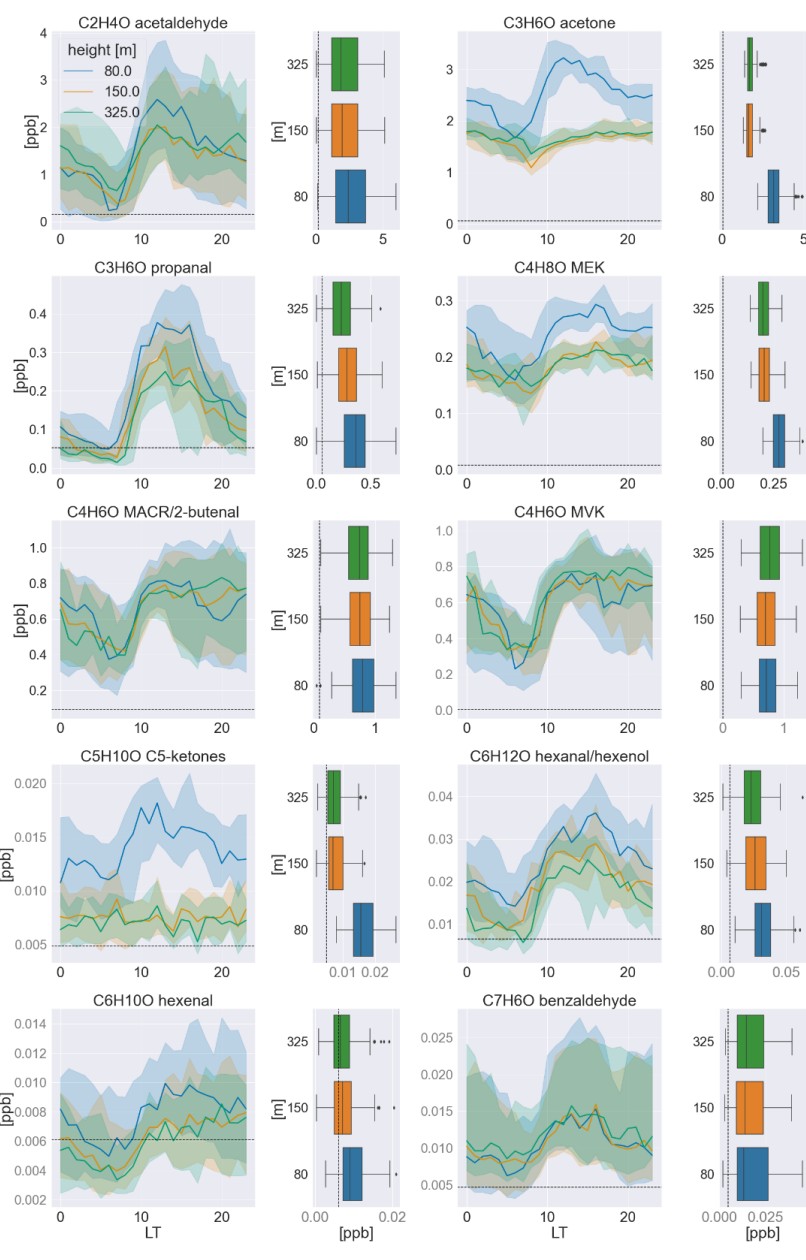

**Figure 2: Median averaged timeseries in the dry season (September/October) of 2019 measured at all sampling heights for each carbonyl compound and its respective vertical profile at noon (12:00–15:00 LT) to the right. The shadings indicate the quartiles (25th and 75th). In the box-and-whisker plots, the boxes also represent the quartiles, while the residual data except for outliers are included in the whiskers. The detection limit (3 sigma) is indicated by dashed, black lines. The mixing ratios in black font were calibrated to a standard, while those in gray font were calculated based on the k-rate.**





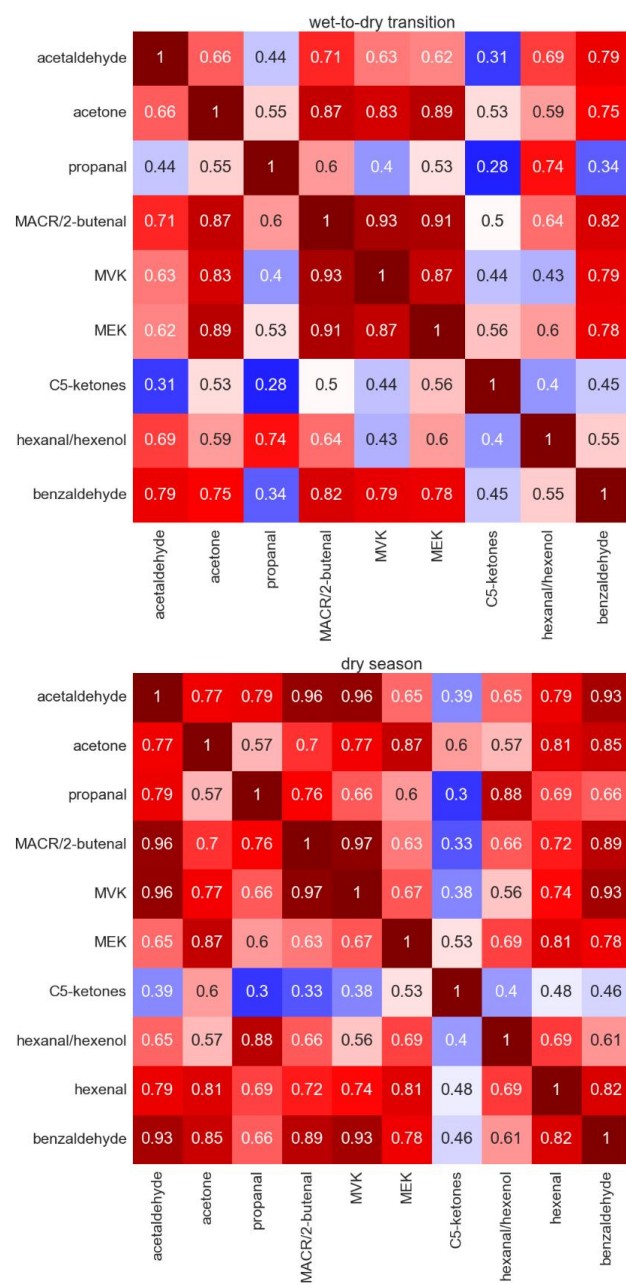


**Figure 3: Pearson correlation coefficients for the intercorrelation of carbonyl species in the wet–to–dry transition season (upper) and in the dry season (lower).**

air masses of varying ages and secondary production and depletion take place simultaneously. When correlating two
time series of BVOC measured at that height, a high correlation coefficient can indicate similar production and loss



or possibly a precursor–product relationship. Here, it is very important to consider the timescales of production and
loss versus vertical transport since the residence time of airmasses in the first 80 m is limited during daytime
convective conditions. In a previous study, the mixing timescale, which accounted for turbulent up and downward
motions between 80 and 325 m at ATTO, was determined to range on average from 60 minutes at 10:00 LT to 15 min
at 15:00 LT (Ringsdorf et al., 2023). Based on that study, we assumed the mean residence time between the canopy
and 80 m to be in the same time range of minutes to 1 hour. Carbonyl precursors including alkenes, isoprene, higher
terpenes, and alkanes have atmospheric lifetimes with respect to oxidation by OH radicals of $\tau > 2$ hours, $\tau \approx 3$ hours,
minutes to hours, and days to weeks for the much less reactive alkanes (Altshuller, 1991; Wolfe et al., 2011). The
lifetimes with respect to OH of carbonyl compounds themselves range from 12 hours (trans-3-hexenal) (Jiménez et
al., 2007) to 119 days (acetone) and are even shorter when considering photolysis, which is a significant sink for
carbonyls (Mellouki et al., 2015). In Table 2, the lifetimes of carbonyls with respect to an average OH concentration
of $1\times10^6$ molecules $cm^{-3}$ based on a previous study at the ATTO site are presented. This is the average over roughly
the same time of day that was considered for carbonyl correlations (10:00–15:00) (Ringsdorf et al., 2023). However,
isoprene oxidation was observed by the daily increase of the product MVK between 80 and 325 m.
Thus, correlations at 80 m will reflect only the processes that occur on a comparable or faster timescale than mixing.
This includes primary emissions, product formation in the atmosphere from short-lived precursors like alkenes and
terpenes, and progressive photochemical degradation/photolysis of short-lived carbonyls as well as loss via deposition.
Figures 3-4 show the Pearson correlation coefficients (p) for both seasons divided into day (10:00–17:00 LT) and
nighttime (22:00–05:00 LT) between the carbonyl compounds and between carbonyls and other selected VOC,
including terpenes (isoprene, sum of monoterpenes), alkenes ($C_5$-alkenes, benzene), and oxygenated compounds
(ethanol, furan, acetic acid, $C_5H_4O_3$). Their diel and vertical distributions are presented in the supplementary figures
S6–S7. $C_5H_4O_3$ is a highly oxygenated compound, which was classified to be exclusively an oxidation product of very
reactive BVOC in a previous study conducted within and above a pine forest. Therein, emission rates of very reactive
BVOC were estimated to reach 6–30 times the emission rates of monoterpenes (Holzinger et al., 2005). In this study,
the highest mixing ratios were found at 325 m, suggesting that besides being formed as a first-order oxidation product
close to the canopy it was also a higher-order oxidation product that therefore emerges at longer timescales. Very
reactive BVOC presumably also represent precursors for carbonyl compounds. Periods with precipitation were
excluded from the correlations to avoid the effects of downbursts and washout. As expected, high correlations were
found between isoprene and the sum of monoterpenes, which are all primary emissions that depend strongly on light
and temperature. Correlation of carbonyl compounds with isoprene and monoterpenes was preferred over PAR and
temperature to identify light- and temperature-dependent direct emission, due to the temporal delay between emission
and detection. However, it is striking that most carbonyls showed significant correlations with all other carbonyls with



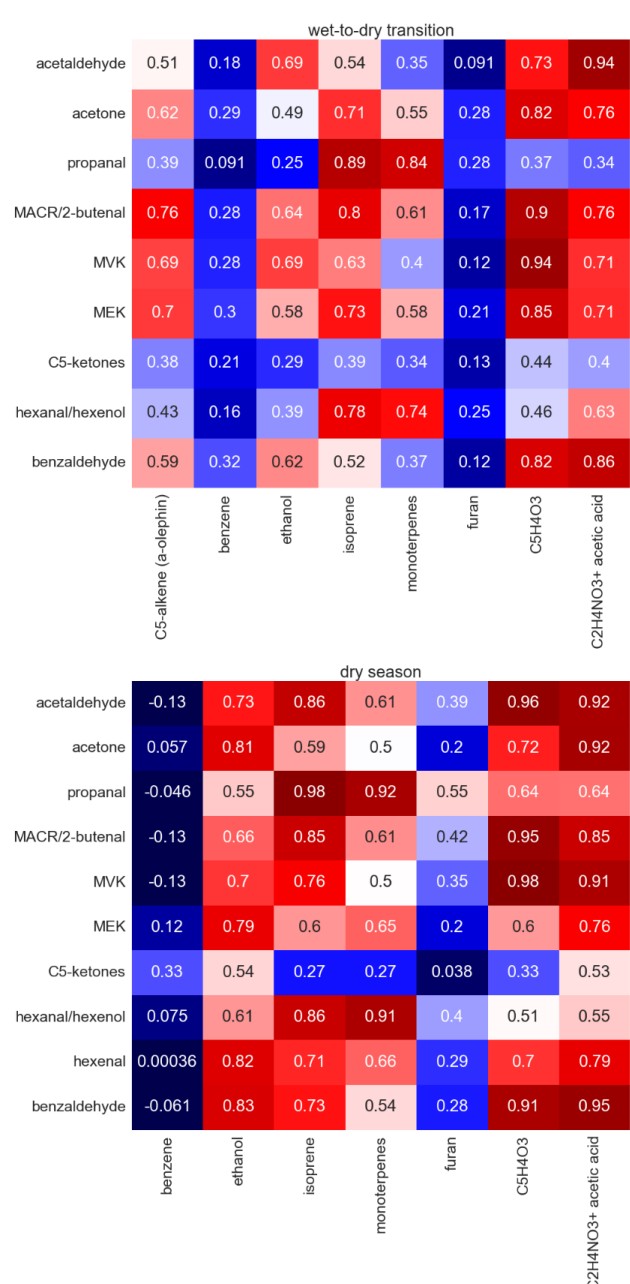

**Figure 4: Pearson correlation coefficients for the correlation of carbonyl species with selected hydrocarbons in the wet–to–dry transition season (upper) and the dry season (lower).**

Pearson coefficients greater than 0.7. This likely resulted from the common driving variables, namely light and
temperature for emission, as well as similar photochemical production rates.



The highest correlations with the primary emitted isoprene and monoterpenes were obtained for propanal and n-hexanal. Other compounds that were found to correlate very well were acetaldehyde, methacrolein, MVK, and $C_5H_4O_3$ as well as acetone and MEK.

## 3.4 Nocturnal loss rates

Biogenic emissions unrelated to photosynthesis might have continued during the night, whereas oxidative chemistry and, thus, secondary production of carbonyls was limited to reactions with $O_3$ and $NO_3$, which were found at low levels in the remote forested atmosphere. Important loss mechanisms at night are deposition to surfaces and reaction with $NO_3$ (Brown and Stutz, 2012). Deposition at night is thought to happen via adsorption to the cuticle of the leaves, since stomata are closed in the absence of light. However, there is evidence that the stomatal conductance is maintained at night by many woody species, implying an irreversible uptake for BVOCs that can be further processed and converted to other metabolites (Niinemets et al., 2014). Reaction with $NO_3$ at nighttime is limited to unsaturated BVOC but is also efficient for some saturated aldehydes. Assuming a rather high mixing ratio of 10 ppt $NO_3$ (Brown and Stutz, 2012; Khan et al., 2015), nighttime atmospheric lifetimes of 8 days are estimated for n-hexanal, the most reactive observed aldehyde with respect to $NO_3$. Ozone is circa 1000 times more abundant than $NO_3$, but the reaction rates with carbonyls are much lower. Therefore, deposition is expected to be the dominant loss mechanism for carbonyls at night. Table 2 summarizes properties of the observed carbonyl compounds that are important driving variables of deposition on surfaces together with their observed loss rates during the night. These rates were obtained by linear regression of the observed nocturnal time series at 80 m between 22:00 and 04:00 LT. Since the uptake of BVOC by leaves occurs only when the ambient concentration exceeds the concentration in the inter-leaf space, high loss rates were observed for BVOC with high ambient mixing ratios. However, the concentration gradient can be maintained by a metabolic transformation of the BVOC in the leaf. Table 2 also includes the reactivity of BVOC towards the OH radical, $O_3$, and $NO_3$.

## 4 Discussion

In the following sections, the diel variability, vertical distribution (80–325 m), and correlations between all measured BVOC are considered in a compound-wise manner. The measurements presented here are related to previous studies on the emission, formation, and loss of the carbonyl species.

## 4.1 Acetaldehyde

Acetaldehyde (ethanal) is known to be an important contributor to the total ambient carbonyl concentration in the atmosphere. Various sources of acetaldehyde have been characterized previously, including direct emissions from vegetation and the ocean and secondary production from the OH-, $NO_3$-, and $O_3$-initiated photooxidation of hydrocarbons (Rottenberger et al., 2004; Wang et al., 2020b). Direct biogenic emissions may be of special importance for the Amazonian rainforest, as acetaldehyde and ethanol release have been reported to result from root anoxia (Bracho-Nunez et al., 2012; Holzinger et al., 2000; Rottenberger et al., 2008). This may occur in large areas caused





**Table 2: Rate coefficients for the reaction with OH, NO₃, and O₃ and the atmospheric lifetime considering an OH radical concentration of 1x10⁶ molecules cm⁻¹. The rate coefficients and boiling point temperature were taken from the NIST database. Water solubility has been reported by Sander et al., 2023. The loss rate is calculated based on the median averaged slopes of the nocturnal (22:00–04:00) carbonyl timeseries.**

| VOC species | $k_{OH}$ [cm³ #⁻¹ s⁻¹] | Estimated lifetime Amazon [days] | $k_{NO3-}$ [cm³ #⁻¹ s⁻¹] | $k_{O3}$ [cm³ #⁻¹ s⁻¹] | Volatility ($T_{boil}$ [K]) | Water-solubility ($H_s^{cp}$) [mol m⁻³ Pa⁻¹] | Loss rate [ppb min⁻¹] (transition, dry season) |
|---|---|---|---|---|---|---|---|
| Acetaldehyde | 1.6E-11 | 1.4 | 2.4E-15 | 3.4E-20 | 294 | 1.3E-1 | -6.7E-4, -1E-3 |
| Acetone | 1.9E-13 | 119.3 | 8.5E-18 | 8.5E-18 | 329 | 2.7E-1 | -7.8E-4, -9.5E-4 |
| Propanal | 2E-11 | 1.2 | 6.2E-15 | - | 322 | 9.9E-2 | -8.9E-5, -1.4E-4 |
| MEK | 1.2E-12 | 19.3 | - | 2.06E-16 | 353 | 1.8E-1 | -8.4E-5, -7.4E-5 |
| MVK | 1.9E-11 | 1.25 | 1.2E-16 | 4.48E-18 | 354 | 4.0E-1 | -1.1E-4, -4.8E-4 |
| MACR | 3E-11 | 0.75 | 3.3E-15 | 1.09E-18 | 341 | 4.5E-2 | -2.1E-4, |
| 2-Butenal | 3.6E-11 | 0.64 | 5.1E-15 | 1.58E-18 | 375.5 | 5.9E-1 | -2.1E-4 |
| 2-pentanone (C₅-ketones) | 4.6E-12 | 5.08 | - | - | 375 | 1.3E-1 | - |
| n-hexanal | 2.8E-11 | 0.83 | 1.1E-14 | - | 402 | 4.5E-2 | -2E-6 |
| Z-3-hexenol | 1E-10 | 0.21 | 2.7E-13 | 6.4E-17 | 427.7 | - | - |
| z-2-hexenal | 4.4E-11 | 0.52 | 1.2E-14 | 2.0E-18 | 419.7 | 1.4E-1 | - |
| Benzaldehyde | 1.3E-11 | 1.78 | 2.01E-15 | 2.0E-19 | 452 | 4.0E-1 | -7.9E-6, -8E-6 |
| Isoprene | 1E-10 | 0.23 | 6.7E-13 | 1.28E-17 | 307 | 1.3E-4 | -1.5E-3, -2.6E-3 |
| α-pinene | 5.3E-11 | 0.43 | 6.2E-12 | 9.6E-17 | 430 | 7E-4 | 1.7E-4, -3.2E-4 |

by seasonal flooding (Parolin et al., 2004). At 80, 150, and 325 m in the wet–to–dry transition season of 2019, observed
mean diurnal concentrations were 642, 702, 852 ppt, respectively, and 1.38, 1.25, 1.47 ppb in the dry season of 2019.
Metabolic production pathways of acetaldehyde within plants and subsequent emission have been found to occur not
only during root-flooding but also during rapid light-dark transitions (Fall, 2003; Holzinger et al., 2000). The anaerobic
conditions at the root's surface during flooding cause the ethanolic fermentation pathway to form ethanol that is
transported to the leaves of the plant to provide an energy source.
Acetaldehyde is an intermediate of this pathway which tends to leak out to the atmosphere due to its high volatility
(Kreuzwieser et al., 2000). Some Amazonian tree species can switch to fermentative metabolism (Bracho-Nunez et
al., 2012), but concentration or flux measurements during the dry–to–wet transition in Amazonia under field
conditions are missing. In this study, a strong correlation was found for ethanol and acetaldehyde in the nighttime
during the transition season (p = 0.92). Ethanol mixing ratios were ten times higher in the transition season and showed
a diel maximum at nighttime. Since river levels were at their maximum levels in the transition season, root flooding





**Table 3: Median averaged mixing ratios of the observed carbonyl compounds for the measurement periods in the dry–to–wet transition and the 2019 dry season. The range presents the lowest mixing ratio included in the 25th and the highest mixing ratio included in the 75th quantile of the median averaged diurnal cycle. Numbers in italics represent the limit of detection.**

| Carbonyl species | Height [m] | Wet-to-dry transition – Jun-Jul 2019 | | Dry season 2019 – Sep-Oct 2019 | |
|---|---|---|---|---|---|
| | | Median [ppt] | Range [ppt] | Median [ppt] | Range [ppt] |
| Acetaldehyde | 80 | 642 | 24 - 2043 | 1380 | *160* - 3179 |
| | 150 | 702 | 24 - 1801 | 1252 | 239 - 3134 |
| | 325 | 852 | 59 - 1883 | 1472 | 256 - 3716 |
| Acetone | 80 | 1333 | 559 - 2083 | 2546 | 1155 - 3812 |
| | 150 | 1124 | 494 - 1509 | 1657 | 918 - 2105 |
| | 325 | 1146 | 661 - 1545 | 1707 | 1087 - 2115 |
| Propanal | 80 | 176 | 71 - 438 | 165 | *53* - 410 |
| | 150 | 150 | 58 - 365 | 115 | *53* - 349 |
| | 325 | 119 | *53* - 348 | 85 | *53* - 305 |
| MEK | 80 | 177 | 82 - 267 | 249 | 116 - 348 |
| | 150 | 175 | 81 - 229 | 188 | 98 - 259 |
| | 325 | 177 | 105 - 240 | 185 | 75 - 247 |
| MVK | 80 | 184 | 45 - 483 | 607 | 100 - 994 |
| | 150 | 229 | 53 - 481 | 599 | 164 - 967 |
| | 325 | 265 | 93 - 499 | 659 | 239 - 1014 |
| MACR/2-Butenal | 80 | 415 | 149 - 755 | 679 | 250 - 1026 |
| | 150 | 425 | 149 - 697 | 644 | 307 - 1010 |
| | 325 | 439 | 181 - 694 | 644 | 281 - 996 |
| $C_5$-ketones | 80 | 11 | *7* - 18 | 14 | *7* - 23 |
| | 150 | 9 | *7* - 15 | 8 | *7* - 13 |
| | 325 | 9 | *7* - 15 | *7* | *7* - 12 |
| n-hexanal/ hexenols | 80 | 15 | *6* - 50 | 26 | *6* - 53 |
| | 150 | 11 | *6* - 33 | 19 | *6* - 42 |
| | 325 | 9 | *6* - 30 | 16 | *6* - 43 |
| z-2-hexenal | 80 | - | < 11 | 8 | *6* - 14 |
| | 150 | - | < 9 | - | < 12 |
| | 325 | - | < 9 | - | < 12 |
| Benzaldehyde | 80 | 12 | *6* - 27 | 11 | *6* - 26 |
| | 150 | 12 | *6* - 24 | 10 | *6* - 26 |
| | 325 | 12 | *6* - 24 | 11 | *6* - 25 |

may be responsible for the seasonal variability of ethanol (Kirstine and Galbally, 2012). However, acetaldehyde
showed a different seasonal variability, indicating other sources were dominant. It is important to keep in mind that
the ATTO site is a "Terra firme" region with inundation events rare. Field measurements of roots under anoxia are
still missing. Fast light–dark transitions occur continuously inside the forest canopy and are suspected to lead to an
overproduction of cytosolic pyruvic acid in the leaves that is converted to acetaldehyde as a safety mechanism against
acidification (Fall, 2003). The wounding of a plant through cutting or drying out of plant tissues also leads to release
of acetaldehyde (Guenther, 2000). The compound is also found in emissions from leaf litter presumably as a byproduct
of biomass degradation (Schade and Goldstein, 2001; Karl et al., 2003). Furthermore, the oxidation of polyunsaturated
fatty acids in leaves leads to the formation of reactive aldehydes, which can represent a primary source for many
aldehydes, including acetaldehyde (Niinemets et al., 2014; Matsui et al., 2010). Once released, the atmospheric
lifetime of acetaldehyde is in the range of 1.4 days with respect to OH (Table 2).



Tree branch enclosures and vertical gradient measurements at another Amazonian measurement site (Rondônia) in
1999 revealed that the canopy's role as a sink can even exceed its function in emissions. Uptake to leaves mainly
occurred via the leaf stomata and has been reported to be governed by a compensation point that varies between
canopy and understory species. The authors concluded that the observed ambient concentrations were generated
mainly by the secondary photochemical production of acetaldehyde (Rottenberger et al., 2004). Accordingly, in 2013,
measurements of acetaldehyde vertical gradients below 80 m at ATTO showed increasing acetaldehyde between 24 m
(inside the canopy, high influence by surrounding trees) and 79 m. However, interestingly, this has only been observed
in the dry season (Sep), whereas during the wet season of 2013 (Feb/Mar), a dominance of primary emission over
secondary production was indicated by decreasing concentrations directly above the canopy (Yáñez-Serrano et al.,
2015). We observed decreasing mixing ratios at altitudes above 80 m under dry season or close to dry season
conditions in 2019. At noon, the acetaldehyde mixing ratios peaked in the first 150 m above the canopy, consistent
with a rapid secondary production and a possible contribution from direct emission. Primary emission might vary in
strength and dominance with season due to the variability of light, temperature, precipitation, and soil moisture and
due to plant phenology. At 150 and 325 m, similar mixing ratios were measured, suggesting well-mixed conditions
and ongoing secondary formation between those heights, due to the many routes of acetaldehyde photochemical
generation.
In the rainforest environment, sources of the photochemical precursor hydrocarbons of acetaldehyde are most likely
to be natural emissions or longer-lived emissions from distant biomass burning. Aldehydes are a common product of
any hydrocarbons that are oxidized in the atmosphere (Mellouki et al., 2015; Calogirou et al., 1999). Laboratory
experiments showed that acetaldehyde emerges from the oxidation of alkanes and alkenes, with ethane and propene
having the largest emission fluxes globally (Singh et al., 2004). Ethane is globally distributed; thus, background
concentrations of acetaldehyde are generated by this route, which are, however, low due to the rapid subsequent
transformation via reaction with OH. Biogenically emitted compounds with high molar yields for the formation of
acetaldehyde are $> C_2$ alkenes (0.85) and ethanol (0.95). Additionally, isoprene and terpenes have a low molar yield
(0.019, 0.025) but exhibit the strongest emissions measured from the forest (Fischer et al., 2014). The reaction of other
aldehydes butanal, 2-pentanone, and 2-heptanone with OH and $NO_3$ also leads to the formation of acetaldehyde,
sometimes with high yields (Atkinson et al., 2000). In the data presented here, acetaldehyde at 80 m correlated best
with photolytically generated species like MVK, methacrolein, and $C_5H_4O_3$ (p = 0.96) and with benzaldehyde
(p = 0.93) in the dry season and correlated well with acetic acid (p > 0.92) in both seasons. In the transition season
correlations were weaker overall (Figs. 3–4), which could hint at different primary and secondary acetaldehyde
sources. Correlation coefficients of acetaldehyde and BC at 325 m were below 0.6 at daytime but at nighttime, in the
transition season, a rather high correlation with p = 0.82 was observed (Fig. S6).
From about 16:00 LT onwards until the next day the vertical gradient is reversed with the lowest concentrations at
80 m. This likely reflects the uptake to plant tissues regulated by compensation points since the $NO_3$ and $O_3$ reactivity
is rather low. Acetaldehyde exhibits the strongest observed loss rate at nighttime among all the carbonyl compounds
in the dry season and it had the highest Henry's law constant (Table 2).
**4.2   Acetone**
Acetone (propanone) is the simplest ketone and the most abundant and widespread OVOC in the atmosphere due to
its relatively long atmospheric lifetime of 15 days (Singh et al., 2004) (primarily driven by photolysis in the upper
troposphere, 119 days with respect to OH). The variation of acetone mixing ratios throughout the day above the
roughness sublayer at 150 and 325 m was small compared to the other carbonyls. However, mixing ratios at 80 m
increased substantially with light and temperature during the day. In the wet–to–dry transition season mixing ratios
reached 1.33, 1.12, 1.15 ppb on average, while in the dry season, 2.55, 1.66, 1.71 ppb (80, 150, 325 m) were measured.
The vertical distribution of acetone showed clearly enhanced mixing ratios at 80 m during daytime compared to well-
mixed conditions at higher altitudes. The strong gradient in the first 150 m above the canopy indicated a large positive
flux above the canopy. Possible reasons for this strong gradient include primary biogenic emission, biomass burning,
production from leaf wax, or efficient secondary formation from very short-lived precursors that exceed deposition
and stomatal uptake. Flux measurements in a tropical forest in Costa Rica in the dry season have found a bidirectional
but net-positive canopy flux of acetone (Karl et al., 2004). Additionally in 2013, when several carbonyls were



measured at ATTO below 80 m, the acetone mixing ratios inside the canopy (influenced by surrounding trees) were
lower than the values measured at 79 m in the dry season. As for acetaldehyde, these increasing vertical gradients
suggested a dominance of photolytically secondary formation over direct emission. In the wet season in 2013,
however, mixing ratios measured in the canopy and at 79 m were of similar magnitude and, compared to the dry
season, much lower at both heights. In conclusion, no clear dominance of secondary formation or direct emission was
found in the wet season (Yáñez-Serrano et al., 2015). We also observed seasonal differences at all three heights, with
lower mixing ratios in the transition compared to the dry season. While we cannot report acetone observations from
the wet season, we did observe higher correlations of acetone with non-primary emitted OVOC methacrolein, MVK,
and $C_5H_4O_3$ in the transition season (p > 0.82) compared to the dry season, suggesting that in 2019 secondary
formation contributed more acetone to observed mixing ratios in the transition season than in the dry season. In light
of the widely differing atmospheric lifetimes of acetone and those OVOC, the most likely explanation for the high
transition season correlations is a dominating secondary acetone source at a similar rate. Contributions from aged
biomass burning plumes containing acetone in the dry season, when enhanced BC concentrations were observed,
could also be the reason for a weaker correlation of $C_5H_4O_3$, methacrolein, and MVK with acetone in the dry season.
Based on the information obtained in 2013 and the data from this study, secondary production in the dry and transition
season appears to peak between the canopy and 150 m above ground, adding up to varying contributions of direct
emissions from vegetation. In conclusion, the most relevant precursors were very reactive biogenic compounds. The
best correlations were found with MEK (p > 0.87) in both seasons, which is another long-lived ketone that is known
to be directly emitted from the Amazon rainforest and produced in the atmosphere overhead (Yáñez-Serrano et al.,
529  2015, 2016).

Primary sources of acetone are direct emission from vegetation and, to a smaller extent, also from dead plant matter.
Acetone is released during cyanogenesis, which acts as a repellant that stops herbivores eating the plant's leaves.
During the production and release of volatile hydrogen cyanide, which deters the feeding herbivore, acetone is formed
as a byproduct. Cyanogenesis occurs in many plant species, though some employ different mechanisms to produce
hydrogen cyanide so that other carbonyl byproducts can be released (Fall, 2003). Another known biogenic pathway
for acetone formation is acetoacetate decarboxylation in soil bacteria and humans (Fall, 2003). Both light and
temperature have been suspected to drive acetone emissions, as shown for some pine and spruce trees (Seco et al.,
537  2007).

Secondary formation of acetone is known to occur from anthropogenically emitted $C_3$-$C_5$ isoalkanes (propane,
isobutane, isopentane) and biogenic emitted methyl butenol and certain terpenes (Seco et al., 2007; Fischer et al.,
2014; Jacob et al., 2002). We found mixing ratios of isopentane to be below the detection limit (13 ppt), and the
vertical distribution and correlations reported for acetone indicated a rapid formation in the first 150 m above ground
by hydrocarbons that are much more short-lived than alkanes.
At night, deposition could be observed on the basis of the rapidly decreasing mixing ratios at 80 m, compared to the
slowly occurring reactions with $NO_3$ and $O_3$. Similar effects have been reported in flux measurements performed by
Karl et al. (2004).

### 4.3 Propanal

Propanal is an isomer of acetone and is not distinguishable from acetone by classical PTR-MS type instruments using
$H_3O^+$. In this study, the first high temporal resolution measurements of propanal in a tropical forest are presented, and
the vertical distribution above the canopy was found to differ markedly from acetone. In general, in the remote
atmosphere, we may expect the more reactive propanal to have lower mixing ratios than acetone, although this may
not be true close to sources. We measured average concentrations of 176, 150, and 119 ppt in the wet–to–dry transition
and 165, 115, and 85 ppt in the dry season (80, 150, 325 m). The ratio of propanal to acetone in the roughness sublayer
of the tropical forest and above yields 1:7.6, 1:9.6 (transition season) and 1:15.4, 1:20 (dry season) at 80 and 325 m.
Globally, the mixing ratio of propanal has been estimated to be about one-third of acetaldehyde (Singh et al., 2004),
while at ATTO a ratio of 1:4.2, 1:8.1 (transition season) and 1:7.2, 1:14 (dry season) was measured at 80 and 325 m,
respectively. However, it should be noted that, globally, a large propanal source is propane oxidation, which is
predominantly emitted from anthropogenic activities associated with oil and gas use. Acetaldehyde sources in the
rainforest thus far exceed propanal sources in the context of the global budget (Warneck, P.; Williams, J., 2012).



Propanal emission from vegetation has been reported for non-tropical forests (Guenther, 2000; Villanueva-Fierro et
al., 2004), although the metabolic pathway was not specified. Wang et al. (2019) described the biosynthesis of
acetaldehyde and propanal during fruit ripening. It was also noted that propanal emission occurs from ferns (Isidorov
et al., 1985), which is important since fern species are common in the understory of tropical forests.
Throughout the day, propanal exhibited a negative vertical gradient (i.e., decreasing mixing ratio with increasing
height). This occurs most likely due to dilution and photochemical loss of propanal generated in or emitted from the
canopy. A similar distribution was also observed for monoterpenes and isoprene, which are primary emitted VOC.
Accordingly, propanal observed at 80 m also correlates best with isoprene in both seasons ($0.89 > p > 0.98$) followed
by monoterpenes ($0.84 > p > 0.92$). The estimated atmospheric lifetime of propanal of about 1 day (Guimbaud et al.,
2007) (1.2 days for the oxidation by OH, table 2) is similar to that of acetaldehyde, but the vertical profiles revealed
different distributions in the first 325 m above ground (Figures 1 and 2). The weak gradient of acetaldehyde between
150 and 325 m at daytime in contrast to the steadily decreasing vertical profile of propanal can thus only be explained
by a higher yield of acetaldehyde from secondary production above 150 m. This is not surprising since acetaldehyde
is produced during oxidative degradation of many hydrocarbons. The secondary production of propanal is known to
occur via the photochemical oxidation of $C_3$ and larger hydrocarbons (Singh et al., 2004) and propane (Altshuller,
1991). Their lifetimes range from 5 days to a few hours (Altshuller, 1991). However, due to the high correlation of
propanal and isoprene, which is even higher than the correlation of isoprene and its oxidation products MVK and
methacrolein, a primary and mainly light-dependent source is surmised.
Nighttime mixing ratios of propanal were decreasing at 80 m (Table 2). Since the reaction rate of propanal with $NO_3$
is faster and the water solubility lower than that of the other carbonyl compounds, a higher fraction could potentially
react in the atmosphere. Stomatal uptake for the terpenes might be driven by the concentration gradient between leaf
and atmosphere, and the same might hold for propanal.
**4.4    Methyl Ethyl Ketone (MEK)**
Mixing ratios of MEK in the wet–to–dry transition were 177, 175, and 177 ppt on average, compared to 249, 188, and
185 ppb in the dry season (80, 150, 325 m). With a conventional PTR-MS, butanal and MEK are detected at the same
exact mass, whereas in this study using $NO^+$ reagent ions solely MEK was measured. Butanal mixing ratios were
determined to be below the detection limit (20 ppt); thus, the contribution of butanal to MEK for PTR-MS can be
assumed to be very low. The mixing ratios obtained in this study agree well with previous studies conducted with a
PTR-quadrupole-MS at the ATTO site in 2013 and close to Manaus (Amazonia) 2014, which would not completely
exclude possible interferences on the nominal mass of MEK (Yáñez-Serrano et al., 2015, 2016). The vertical
distribution of MEK throughout the day resembles that of acetone in both seasons. Mixing ratios above the roughness
layer (at 150 and 325 m) were almost uniform, while those at 80 m showed a more pronounced diurnal cycle with
strongly increasing values in the day and decreasing values at night. As well as being structurally similar to acetone,
MEK also has a long lifetime of 4.3 days (Fischer et al., 2014) (19.3 with respect to OH oxidation alone) relative to
mixing timescales. MEK is also known to have primary and secondary sources (Yáñez-Serrano et al., 2016).
Therefore, it is not surprising that MEK correlated best with acetone at 80 m ($p = 0.87$ in the dry season), but in the
transition season, it also correlated well with $C_5H_4O_3$, methacrolein, and MVK. This suggests secondary sources from
biogenically emitted precursors were more dominant during the transition season than in the dry season, similar to
acetone.
MEK emissions have been reported for rainforest canopies (Yáñez-Serrano et al., 2015) and fern (Isidorov et al.,
1985), decaying plant matter (Warneke et al., 1999), fungi, and bacteria (Yáñez-Serrano et al., 2016). The metabolic
pathways of production and the release mechanisms are poorly understood but have been suggested to involve plant
signaling, injured leaves, and root-aphid interactions (Yáñez-Serrano et al., 2016). Within-plant conversion of the
cytotoxic 1,2-ISOPOOH, which was deposited on poplar leaves, first to MVK and subsequently to MEK, has been
reported to represent a large biogenic source of MEK. The enzyme responsible for the conversion of MVK to MEK
is widespread among plants (Canaval et al., 2020).
The secondary formation of MEK occurs via the oxidation of n-butane with a yield of 80% (Singh et al., 2004) and
via oxidation of 2-butanol, 3-methyl pentane, and 2-methyl-1-butene (Yáñez-Serrano et al., 2016). Additionally, all



alkenes with a methyl/ethyl group on the same side of the olefin bond are possible precursors of MEK (Singh et al.,
2004). Butane was not expected to be abundant in the rainforest environment due to its anthropogenic sources, and
butane oxidation would also yield butanal, which was only detected below the LOD. As for acetone, the vertical
distribution and correlations discussed above suggest higher levels of short-lived precursors of MEK than of alkanes.
Rapidly decreasing concentrations at 80 m at night are in agreement with earlier studies, that reported deposition of
MEK in the canopy due to its high water solubility (Yáñez-Serrano et al., 2016) (Table 2).

### 4.5   Methyl Vinyl Ketone and Methacrolein/2-Butenal

The main source of both carbonyls MVK and methacrolein is the oxidation of isoprene by OH. Thus, they are
summarized in one section. It has been shown previously that methacrolein is detected together with 2-butenal (Koss
et al., 2016), which is also found in vegetation emission studies, albeit in small concentrations (Isidorov et al., 1985;
Hellén et al., 2004). (E)-2-butenal is a signaling compound within the plant that serves to trigger responses to abiotic
stress (Yamauchi et al., 2015). Its atmospheric lifetime is around 20 hours, slightly longer than the lifetimes of
methacrolein and MVK which are10 and 14 hours, respectively (Hellén et al., 2004; Liu et al., 2016). It is also known
that MVK and methacrolein cannot be detected separately from isoprene hydroxyhydroperoxides (ISOPOOH) without
using a scrubber, since the hydroperoxides decompose onto the same m/z. With $NO^+$ CIMS the fragment of 1,2-
ISOPOOH and methacrolein share one m/z-ratio, while 4,3-ISOPOOH is detected together with MVK (Rivera-Rios
et al., 2014). Wall exchange effects in the inlet line might have led to the removal of ISOPOOH from the sampled air
due to their reduced volatility, but a contribution to the MVK and methacrolein signal remains possible. ISOPOOH
also originate from the oxidation of isoprene and are very reactive, reflected by lifetimes of 3 and 2 hours. After the
initial reaction of OH and isoprene, the subsequently formed peroxy radical ($RO_2$) can react with NO to form MVK
and methacrolein, but it can also react with $HO_2$ to form ISOPOOH (Liu et al., 2016). At close to pristine conditions
at ATTO, NO mixing ratios are low, and the yield distribution between ISOPOOH, MVK, and methacrolein was
estimated to be 50, 25, and 25%, respectively (Rivera-Rios et al., 2014; Ringsdorf et al., 2023). It has been shown that
the oxidation of isoprene can proceed already within plant tissues by reaction with accumulated reactive oxygen
species. The accumulation of reactive oxygen species, including OH, is a reaction to biotic and abiotic stresses and
can exceed the antioxidant defense capacities in the tissue. The oxidation of isoprene within the tissue reduces the
amount of reactive oxidized species and leads to the direct emission of isoprene's products MVK and methacrolein,
especially under stress (Jardine et al., 2012b, 2013).
Oxidation of the monoterpene ocimene has been identified as another secondary source for MVK (Calogirou et al.,
1999). To our knowledge, there are no other significant direct or secondary sources of MVK, methacrolein, and
ISOPOOH other than the oxidation of isoprene. This explains the observed diurnal cycle with an afternoon maximum
due to the light-dependent emission of isoprene and the photochemical production of OH. Since isoprene is present at
relatively high mixing ratios at all tree sampling heights (3.69, 3.33, 3.0 ppb at 80, 150, and 325 m in the dry season),
the oxidative formation of MVK, methacrolein, and ISOPOOH takes place throughout the mixed layer. The observed
slightly increasing mixing ratios of MVK with height are consistent with rapid isoprene oxidation above the canopy,
slower removal of MVK itself, and turbulent in-mixing of cleaner air from above. Isoprene has an estimated
atmospheric lifetime of about 3 hours, and it was previously reported that only circa 10% of emitted isoprene was
oxidized within the canopy (Karl et al., 2004). Unlike MVK, methacrolein and 2-butenal show a slightly decreasing
vertical gradient. Sources and sinks of MVK and methacrolein are very closely related, so the presence of significant
quantities of 2-butenal is the most likely explanation for that difference.
Dry deposition to leaf surfaces has been observed in a previous study for the sum of MVK and methacrolein and
individually for these compounds during daytime (Nguyen et al., 2015; Tani et al., 2010). Uptake by leaves represents
a significant sink that exceeds loss via OH oxidation near leaves (Tani et al., 2010). In this study, the rapid decrease
of nocturnal concentrations at 80 m indicated that deposition at night was also taking place.
MVK and methacrolein + 2-butenal showed similar mixing ratios in the dry season of 607, 599, 659 ppt MVK and
679, 644, 644 ppt methacrolein + 2-butenal. It has to be considered that the uncertainty of MVK mixing ratios is higher
than the uncertainty of methacrolein mixing ratios due to their k-rate-based calculation rather than calibration to a gas
standard. In the transition season, methacrolein + 2-butenal (415, 425, 439 ppt) exceeded the mixing ratios of MVK



(184, 229, 265 ppt). Whether that resulted from the high seasonal variability of 2-butanal or from the contribution of
ISOPOOH, unfortunately, remains unclear. Lower levels of all isoprene oxidation products were expected as a result
of lower isoprene mixing ratios and photo-oxidation rates in the transition season.

### 4.6   Sum of $C_5$-ketones

The mixing ratios obtained for the sum of $C_5$-ketones were 11, 9, and 9 ppt in the transition season, while slightly
higher levels of 14, 8, and 7 ppt (80, 150, 325 m) were obtained in the dry season. A diurnal cycle was observed at
80 m only, whereas levels at 150 and 325 m were similar and showed no trend throughout the day and night. $C_5$-
ketones were 2- and 3- pentanone as well as 3-methyl-2-butanone. The atmospheric lifetime of 2-pentanone is in the
range of 5 days. All three ketones have been included in emission inventories from plants (Isidorov et al., 1985; König
et al., 1995; Kesselmeier and Staudt, 1999), but there is little information on metabolic pathways or mechanisms. 2-
pentanone has been identified as a marker for fungal activity in indoor environments (Kalalian et al., 2020), since it is
produced in the hyphae of *Aspergillus niger* (Lewis, 1970), a fungus that was also found to degrade biomass in the
Amazon. 3-pentanone is one of the $C_5$ green leaf volatiles (GLV) emitted at lower rates than $C_6$ GLV, which are
described in the next section (Jardine et al., 2012a). An increase of 3-pentanone coincident with high temperatures
after noon was observed at another measurement station in the Amazon rainforest, with a simultaneous decrease of
terpenoid emissions (Jardine et al., 2015). Consistent with this observation, in this study, the correlation of $C_5$ ketones
with isoprene or monoterpenes was low in the transition and dry season during the daytime ($p < 0.39$). The best
correlations for $C_5$-ketones of $0.53 < p < 0.6$ were obtained with acetone and MEK. This was most likely a
consequence of common sources, including primary emission and formation from rather short-lived hydrocarbons and
of the long atmospheric lifetimes relative to mixing timescales, which the observed ketones have in common. Above
150 m, no diurnal variability was observed, which is also in agreement with the other ketones, suggesting they were
well-mixed above the Amazonian roughness sublayer.
Fumigation experiments with different VOC have shown a loss of all three $C_5$-ketones on leaf surfaces (Tani and
Hewitt, 2009). A decrease of the mixing ratios at 80 m could be observed at nighttime, and a high water solubility of
the ketones indicated a high loss rate. However, the signal was too noisy to determine loss rates from the data.
$C_5$-aldehydes, which were usually detected together with the $C_5$-ketones, exhibited lower mixing ratios, especially in
the dry season. Overall, the mixing ratios were below their LOD and thus not investigated in detail. However, a diurnal
and vertical pattern of $C_5$-aldehydes with vertical and diurnal variabilities different to those of the $C_5$-ketones was still
apparent.

### 4.7   n-Hexanal/Hexenols and Hexenals

$C_6$-aldehydes, namely n-hexanal, Z-2-hexenal, Z-3-hexenal, E-2-hexenal, and E-3-hexenal, together with $C_6$-alcohols
and esters form a group that is often termed green leaf volatiles (GLV). Although different temporal variabilities were
observed for n-hexanal/hexenols and hexenals, we here discuss them together in one section due to their common
sources.
In the chloroplasts of almost all green plants, GLV are synthesized from fatty acids as part of the oxylipin pathway.
Their emission results from wounding or mechanical damage, from abiotic factors (such as wind), herbivores, and
pathogen attack (Scala et al., 2013). The amount of GLV emitted from corn plants has been shown to depend on soil
water content, temperature, light, and fertilization, with a stronger emission response at higher temperatures
(Gouinguené and Turlings, 2002). Furthermore, emission has been reported as a response to abiotic stress from light–
dark transitions (Jardine et al., 2012a). Their production and release can be very fast; in the case of Z-3-hexenal,
emission begins only 1 or 2 seconds after damage (Fall et al., 1999). On one hand, GLV have antibiotic properties and
protect the wounded tissue from invading bacteria or other microorganisms. On the other hand, their rapid production
and release make them useful for intra and inter-chemical communication in plants, for example for priming defense
mechanisms. It has been found that a herbivore-infested plant releases signaling compounds, like GLV to attract the
predator (insects, beetles, birds, etc.) of the herbivore (Scala et al., 2013; Mumm and Dicke, 2010; Zannoni et al.,
2020a). The release of GLV can happen on short timescales of minutes to hours but in cases of repetitive wounding



or drying of leaves, the emission can be continuous over days (Scala et al., 2013; Fall et al., 1999). Release of GLV is also caused by drought stress, and GLV levels have been observed to increase at noon as a result of high temperatures in the Amazon forest (Jardine et al., 2015; Kesselmeier and Staudt, 1999).

It remains unclear if the leaf alcohol Z-3-hexanol contributed to the hexanal signal. Z-3-hexanol is also a GLV and has been reported to represent a major part of the emission of many studied plants (Kesselmeier and Staudt, 1999). Its atmospheric lifetime was calculated to be 5 hours with respect to OH. Further, the less abundant isomers, such as Z-4-hexenol or E-2-hexenol, are also likely to contribute to the hexanal signal. Due to photolysis and reaction with OH, the lifetime of n-hexanal is about 4 hours (12 hours for oxidation by OH only), which is also true for E-2-hexenal (Jiménez et al., 2007). Z-3-hexenal has a shorter atmospheric lifetime of 3 hours (Xing et al., 2012).

At ATTO, the integrated emissions from a large uniform area were measured, which made it impossible to detect single wounding events, except for large-scale storm damage or human activities such as forest clearing. Measured mixing ratios were 15, 11, and 9 ppt for n-hexanal in the transition season and 26, 19, and 16 ppt in the dry season. Hexenals were detected at mixing ratios below LOD (6 ppt) for most parts of the day in the transition season, and 8 ppt were measured at 80 m in the dry season. Nighttime mixing ratios of hexenals at 150 and 325 m were, however, also below the LOD (6 ppt). During both measurement phases, n-hexanal was continuously present, exhibiting a distinct diurnal cycle with maximum mixing ratios in the afternoon and higher values in the dry season. Since the emission rate of damaged leaves of hexenals was found to be higher compared to n-hexanal (Fall et al., 1999), the contribution of hexenols to the signal of n-hexanal was very likely. Average daytime mixing ratios between 40 and 70 ppt have also been observed for hexanal and/or hexenols in an elevated position above the rainforest of Malaysia (Langford et al., 2010). To investigate whether the diurnal cycle results from temperature-dependent emission of GLV or additional secondary formation, measurements inside the canopy are required.

It was not surprising that a continuous decrease in both n-hexanal and hexenols with height was observed throughout the day, similar to propanal and other reactive primary emissions like isoprene and monoterpenes. Correlations at 80 m with isoprene ($0.78 < p < 0.86$), monoterpenes ($0.74 < p < 0.91$), and propanal ($p = 0.88$, dry season) indicated light- or temperature-driven emission or rapid secondary formation close to the canopy. This correlation is interesting since GLV emissions upon biotic-induced stresses such as herbivory do not necessarily follow a diel cycle. However, boundary layer dynamics might have modulated the diel cycle since mixing between the canopy and atmosphere is most efficient during daytime convective conditions. Additionally, temperature-related drying of leaves could have led to the observed diel variability.

In contrast to n-hexanal/hexenols, hexenals exhibit a more pronounced seasonal variability, with very low mixing ratios, mostly below the LOD, in the transition season. The correlation with isoprene and monoterpenes during the daytime in the dry season was rather low ($p = 0.71$), with the highest correlations for acetone, MEK, benzaldehyde, and ethanol ($p > 0.8$), suggesting primary and secondary sources of hexenals.

For all $C_6$-aldehydes investigated in this section, decreasing concentrations during nighttime at all three heights were observed in the dry season, when $C_6$-aldehydes mixing ratios were generally higher than in the transition season. A slightly slower decrease of 80-m mixing ratios compared to the higher levels in the dry season may indicate a continued nocturnal emission of GLV, which is plausible since production and release from mechanical wounding, stress, or herbivory is possible without light (He et al., 2021).

**4.8 Benzaldehyde**

The average mixing ratios of benzaldehyde measured in this study are 12, 12, 12 ppt in the transition season and 11, 10, 11 ppt (80, 150, 325 m) in the dry season. No seasonal variability or vertical gradient was observed between the measurement periods.

Benzaldehyde is the lightest monoaromatic aldehyde and is formed via the oxidation of other aromatic compounds. It is a major intermediate product of the oxidation of benzyl radicals via OH and, thereby, of all alkyl-substituted aromatic hydrocarbons (Sebbar et al., 2011). Biogenic aromatics, such as volatile benzenoids or larger molecules like lignols, are produced via the shikimate pathway by plants, which is an important metabolic process, but benzaldehyde can also be emitted by microorganisms (Ladino-Orjuela et al., 2016; Laothawornkitkul et al., 2009). The oxidation of



toluene, which has previously been observed to be emitted from forested environments and farm crops (Heiden et al.,
1999), yields benzaldehyde as a product (6-%) (Atkinson and Arey, 2003). Benzaldehyde is also a benzyl alcohol
oxidation product, which has been reported previously to be emitted from biogenic sources (Bernard et al., 2013).
Benzaldehyde is very reactive, with a calculated atmospheric lifetime primarily determined by its photolysis rate of
2.4 hours (Cabrera-Perez et al., 2016) (1.7 days with respect to OH).
Primary emission of benzaldehyde from vegetation has been reported for grass (Kirstine et al., 1998) and elevated
concentrations under and within the canopy of the Amazon rainforest were measured (Kesselmeier et al., 2000). The
high mixing ratios (about 300 ppt) found at the ground were suspected to result from the decomposition of biomass,
specifically the decomposition of lignols within the litter. In that study, the mixing ratios above the canopy were much
lower than those measured at ground level.
The apparent light or temperature-driven diurnal cycle of benzaldehyde suggests secondary photochemical production
from aromatic hydrocarbons, as the shikimate pathway is independent of light (Jan et al., 2021). The atmospheric
lifetime of precursor aromatics ranges from days to weeks (Altshuller, 1991). Secondary production from long-lived
precursors is a feasible explanation for the missing vertical variability of the very reactive benzaldehyde in the first
325 m of the mixed layer. The rather slow secondary production throughout the mixed layer possibly compensated
for the expected loss through oxidation and dilutive mixing. Mixing ratios observed at 80 m were only slightly more
abundant in the dry season compared to higher altitudes, which could mean a stronger contribution of benzaldehyde
emissions. However, the narrow vertical benzaldehyde distribution points towards well-mixed aromatic precursor
hydrocarbons. Daytime mixing ratios of carbonyls that are suspected to be formed predominantly due to
photochemical formation, namely, acetic acid, $C_5H_4O_3$, methacrolein, MVK, but also acetaldehyde and acetone,
correlate very well with benzaldehyde in the dry season ($0.85 > p > 0.95$). In the transition season, the correlation with
the same compounds is smaller ($0.75 > p > 0.86$). Possible explanations for this difference most likely lie in altered
sources of precursors or benzaldehyde itself due to differences in, e.g., litter decomposing activities. It is important to
note that the missing vertical variability could also be a sign of contamination from the measurement tower itself, e.g.,
through temperature-dependent outgassing of its coating. However, the measurement of the fresh paint did not show
elevated benzaldehyde, while the fresh anticorrosion agent emits some benzaldehyde, but at much lower rates than
other VOC, e.g., toluene or xylene.
Globally, dry deposition constitutes a small sink of benzaldehyde in the same range as oxidation by $NO_3$ (Cabrera-
Perez et al., 2016). We observed decreasing mixing ratios at all three heights throughout the night (Table 2). Wet
deposition and uptake to leaves and soil might have been the dominant sink.
There is evidence that benzaldehyde PAN can emerge when transported to high NOx regions (Caralp et al., 1999).
Mixing ratios of PAN are quite high so this must be considered, but photochemical PAN creation potential is the
lowest of the whole group of organic compounds (Derwent et al., 1998).

**5    Conclusion**
In this study, a PTR-ToF-MS was operated using $NO^+$ as the reagent ion for measuring specific carbonyl compounds
at three heights (80, 150, 325 m), in two seasons, and over 24-hour cycles, on the ATTO tower located in the Brazilian
Amazon rainforest. With the more commonly used ionization method for PTR-MS involving $H_3O^+$ ions, aldehyde and
ketone isomers were detected together at the same exact mass. This precludes the investigation of the individual
species. For the first time, mixing ratios of biogenic aldehydes and ketones measured at high frequency are reported
for a rainforest ecosystem. Generally, higher mixing ratios were found in the dry season. To some extent, this can be
attributed to higher temperatures and enhanced light conditions, which drive emissions and photochemical activity.
However, since temperature and PAR were only slightly enhanced in the dry season compared to the wet–to–dry
transition, other aspects such as phenology (gross ecosystem productivity peaking in the dry season) and contribution
of long-lived species from aged biomass burning plumes are of importance. Ketones have atmospheric lifetimes (days
to weeks) that are much longer than the vertical mixing times (15–60 min) (Ringsdorf et al., 2023). Such compounds



can, therefore, be expected to be also present above the lowermost mixing layer (ABL) in the residual layer and free troposphere. Interestingly, elevated ketone mixing ratios in the roughness sublayer observed at 80 m by day suggest a large source at canopy level or just above. At night, the loss of these species indicates a rapid deposition to the canopy or the underlying forest floor. The correlations shown in Figures 3–4 reveal seasonal differences in the partitioning of primary emission from the canopy and the rate of rapid secondary production above the canopy. The most abundant individually measured carbonyls in this study were acetaldehyde and acetone, both effective PAN precursors, followed by isoprene oxidation products and propanal. Note that formaldehyde was not detected by the applied method. The shorter-lived, longer-chain aldehydes observed in this study showed great variation, exhibiting both increasing and decreasing vertical gradients that vary considerably in strength. All carbonyl compounds showed a distinct diurnal cycle which followed the evolution of light and temperature during the day and, for most compounds, a decrease during the night driven in part by reaction with $NO_3$ but more importantly by deposition to plant tissues, as has been shown by flux measurements for a few oxygenated species before (Karl et al., 2004). The nocturnal uptake of these carbonyl compounds is an important aspect of their local-to-regional-scale budget. Based on this data, we hypothesize that the ecosystem can more efficiently produce reduced species such as isoprene and monoterpenes but more efficiently utilize the oxygenated products of these precursors. The importance of uptake followed by metabolization or storage, especially for oxygenated BVOC has been stressed already in the context of bidirectional exchange of BVOC by Niinemets et al. (2014). This would imply that the rainforest exploits atmospheric oxidation to convert products into more useful, metabolizable forms. Similar preferences for the uptake of oxygenated species over terpenes have been reported for epiphytes such as lichen and moss (Edtbauer et al., 2021). This idea can serve as the basic hypothesis for future plant experiments and the observed loss rates of carbonyl species can help to constrain turbulence resolving canopy exchange models. Overall, we need to improve our understanding of the complexity of biological production and consumption and invest into investigations of primary emissions on a leaf or branch level.

Butanal, and carbonyls higher than $C_7$ were found to be minor components of the rainforest atmosphere, as were the alkanes isopentane, methylcyclopentane, sum of 2- and 3-methylpentane and $C_7$ cyclic alkanes. The ratio of the aldehydes propanal and acetaldehyde, which have comparable atmospheric lifetimes and which were shown to correlate very well in previous studies, was found to be much higher with 1:4.2 and 1:7.2 in the transition and dry season at 80 m compared to the global average ratio of 1:3 (Singh et al., 2004), due to the overwhelming predominance of biogenic sources and precursors in the rainforest.

This application of the $NO^+$ CIMS method has enabled the study of the individual carbonyls not accessible using the $H_3O^+$-based method. We, therefore, recommend periodic switching of the reagents to allow for more specific detection of biogenic emissions. This would complement long-term measurements conducted using the $H_3O^+$ ionization method.

**Code availability.** The python code can be provided upon request.

**Data availability.** BVOC data are available on the ATTO data portal (https://www.attodata.org/), a DOI is requested and will follow soon. Meteorological data conducted at the ATTO tower (320 m) in 2019 are available via https://doi.org/10.17871/atto.95.12.742.

**Supplement link**: A link to the supplement will be included by Copernicus, if applicable.

**Author contributions**: AR and AE conducted the BVOC measurements and AR analyzed this data and drafted the manuscript. BH and CP provided the black carbon observations and meteorological parameters conducted at the 325–m–tall tower. MOS and AA conducted the measurements of the meteorological parameters at the 80 m tower. J.W. supervised this study. J.L. supervised the research that led to this study.

**Competing interests**: The authors declare that they have no conflict of interest.

**Acknowledgements:** We acknowledge the support by the German Federal Ministry of Education and Research (BMBF contract 01LB1001A and 01LK1602B) and the Brazilian Ministério da Ciência, Tecnologia e Inovação



(MCTI/FINEP contract 01.11.01248.00) as well as the Amazon State University (UEA), FAPESP, CNPq, FAPEAM,
LBA/INPA, and SDS/CEUC/RDS-Uatumã. We thank Thomas Klüpfel for his help with VOC measurements. We
especially acknowledge the technical and logistical support of the ATTO team (in particular Reiner Ditz and Hermes
Braga Xavier). We also thank Andrew Crozier for creating and providing a detailed map of the ATTO site.

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
