# Peer review of "Investigating Carbonyl Compounds above the Amazon Rainforest 1"

_EGUsphere, 2024_

## Author Comment (AC1)

**Supplementary material**

**Table S1: Product ion distributions for ionization with NO$^+$. Values were obtained from the single compound headspace analysis conducted with the PTR-ToF-8000. Values in bold represent the main m/z used for the measurements in the Amazon rainforest.**

| Carbonyl species | E/N 70 Td | | | | E/N 120 Td | | | |
| --- | --- | --- | --- | --- | --- | --- | --- | --- |
| | Peaking masses | Max. ncps | Relative counts | Formular | Peaking masses | Max. ncps | Relative counts | Formular |
| Acetone | **88.0393** | 13609 | 0.76 | C$_3$H$_6$NO$_2^+$ | **88.0393** | 585 | 0.40 | C$_3$H$_6$NO$_2^+$ |
| | 59.0461 | 3016 | 0.17 | C$_3$H$_7$O$^+$ | 59.0491 | 461 | 0.32 | C$_3$H$_7$O$^+$ |
| | 77.0597 | 745 | 0.04 | C$_3$H$_9$O$_2^+$ | 43.0178 | 405 | 0.28 | C$_2$H$_3$O$^+$ |
| | 43.0178 | 507 | 0.03 | C$_2$H$_3$O$^+$ | | | | |
| Hexanal | **99.0804** | 1989 | 0.79 | C$_6$H$_{11}$O$^+$ | **71.0855** | 572 | 0.38 | C$_5$H$_{11}^+$ |
| | 117.091 | 157 | 0.06 | C$_6$H$_{13}$O$_2^+$ | 99.0804 | 526 | 0.35 | C$_6$H$_{11}$O$^+$ |
| | 100.076 | 152 | 0.06 | C$_5$H$_{10}$NO$^+$ | 43.0542 | 309 | 0.21 | C$_3$H$_7^+$ |
| | 71.0855 | 81 | 0.03 | C$_5$H$_{11}^+$ | 81.0699 | 52 | 0.03 | |
| | 101.0961 | 80 | 0.03 | C$_6$H$_{13}$O$^+$ | 41.0383 | 45 | 0.03 | |
| | 135.114 | 72 | 0.03 | | | | | |
| Benzaldehyde | **105.033** | 114 | 0.93 | C$_7$H$_5$O$^+$ | **105.033** | 91 | 0.91 | C$_7$H$_5$O$^+$ |
| | 99.0804 | 8 | 0.07 | | 99.0804 | 9 | 0.09 | |
| Pentanal | **85.0648** | 2704 | 0.84 | C$_5$H$_9$O$^+$ | 57.0699 | 1048 | 0.64 | C$_4$H$_9^+$ |
| | 86.0726 | 159 | 0.05 | C$_5$H$_{10}$O$^+$ | **85.0648** | 451 | 0.28 | C$_5$H$_9$O$^+$ |
| | 103.075 | 154 | 0.05 | C$_5$H$_{11}$O$_2^+$ | 58.076 | 53 | 0.03 | |
| | 57.0699 | 106 | 0.03 | C$_4$H$_9^+$ | 41.038 | 43 | 0.03 | C$_3$H$_5^+$ |
| | 87.0804 | 99 | 0.03 | C$_5$H$_{11}$O$^+$ | 69.0699 | 45 | 0.03 | |
| Nonanal | 85.0648 | 218 | 0.78 | C$_5$H$_9$O$^+$ | 57.0699 | 40 | 0.51 | C$_4$H$_9^+$ |
| | 88.0393 | 19 | 0.07 | C$_3$H$_6$NO$_2^+$ | 85.0648 | 29 | 0.37 | C$_5$H$_9$O$^+$ |
| | 103.075 | 14 | 0.05 | C$_5$H$_{11}$O$_2^+$ | **141.1274** | 6 | 0.08 | C$_9$H$_{17}$O$^+$ |
| | 57.0699 | 11 | 0.04 | C$_4$H$_9^+$ | 86.07262 | 4 | 0.05 | C$_5$H$_{10}$O$^+$ |
| | **141.1274** | 8 | 0.03 | C$_9$H$_{17}$O$^+$ | | | | |
| | 121.0968 | 8 | 0.03 | | | | | |
| Octanal | **127.1117** | 79 | 0.65 | C$_8$H$_{15}$O$^+$ | **127.1117** | 58 | 0.56 | C$_8$H$_{15}$O$^+$ |
| | 85.0648 | 42 | 0.35 | C$_5$H$_9$O$^+$ | 57.0699 | 39 | 0.38 | C$_4$H$_9^+$ |
| | | | | | 85.0648 | 7 | 0.07 | C$_5$H$_9$O$^+$ |
| Trans-2-hexenal | **97.0672** | 354 | 0.73 | C$_6$H$_9$O$^+$ | **97.0672** | 1134 | 0.76 | C$_6$H$_9$O$^+$ |
| | 128.0768 | 87 | 0.18 | C$_6$H$_{10}$NO$_2^+$ | 55.039 | 203 | 0.14 | |
| | 99.0804 | 27 | 0.06 | C$_6$H$_{11}$O$^+$ | 98.060 | 124 | 0.08 | C$_6$H$_{10}$O$^+$ |
| | 85.0648 | 14 | 0.03 | C$_5$H$_9$O$^+$ | 99.0804 | 40 | 0.03 | C$_6$H$_{11}$O$^+$ |

| Carbonyl species | E/N 70 Td | | | | E/N 120 Td | | | |
|---|---|---|---|---|---|---|---|---|
| | Peaking masses | Max. ncps | Relative counts | Formular | Peaking masses | Max. ncps | Relative counts | Formular |
| Pentanone | **116.0706** | 2125 | 0.95 | $C_5H_{10}NO_2^+$ | **116.0706** | 578 | 0.57 | $C_5H_{10}NO_2^+$ |
| | 87.08044 | 101 | 0.05 | $C_5H_{11}O^+$ | 86.07262 | 109 | 0.11 | $C_5H_{10}O^+$ |
| | | | | | 43.01784 | 89 | 0.09 | $C_2H_3O^+$ |
| | | | | | 58.04132 | 87 | 0.09 | $C_3H_6O^+$ |
| | | | | | 87.08044 | 82 | 0.08 | $C_5H_{11}O^+$ |
| | | | | | 71.04914 | 70 | 0.07 | $C_4H_7O^+$ |
| Methacrolein | **69.03349** | 3114 | 0.58 | $C_4H_5O^+$ | **69.03349** | 586 | 0.45 | $C_4H_5O^+$ |
| | 100.039 | 1528 | 0.29 | $C_4H_6NO_2^+$ | 41.0383 | 562 | 0.43 | $C_3H_5^+$ |
| | 57.03349 | 527 | 0.10 | $C_3H_5O^+$ | 71.04914 | 66 | 0.05 | $C_4H_7O^+$ |
| | 71.04914 | 179 | 0.03 | $C_4H_7O^+$ | 57.03349 | 54 | 0.04 | $C_3H_5O^+$ |
| | | | | | 100.039 | 48 | 0.04 | $C_4H_6NO2^+$ |
| MVK | **100.039** | 3067 | 0.91 | $C_4H_6NO_2^+$ | **100.039** | 240 | 0.69 | $C_4H_6NO_2^+$ |
| | 71.04914 | 319 | 0.09 | $C_4H_7O^+$ | 71.04914 | 94 | 0.27 | $C_4H_7O^+$ |
| | | | | | 43.01784 | 16 | 0.05 | $C_2H_3O^+$ |
| MEK | **102.055** | 7073 | 0.90 | $C_4H_8NO_2^+$ | **102.055** | 985 | 0.64 | $C_4H_8NO_2^+$ |
| | 73.069 | 531 | 0.07 | $C_4H_9O^+$ | 73.069 | 181 | 0.12 | $C_4H_9O^+$ |
| | 57.03349 | 232 | 0.03 | $C_3H_5O+$ | 57.03349 | 158 | 0.10 | $C_3H_5O^+$ |
| | | | | | 43.01784 | 155 | 0.10 | $C_2H_3O^+$ |
| | | | | | 72.05697 | 71 | 0.05 | $C_4H_8O^+$ |
| Butanal | **71.04914** | 2453 | 0.89 | $C_4H_7O^+$ | 43.05423 | 434 | 0.63 | $C_3H_7^+$ |
| | 72.05697 | 139 | 0.05 | $C_4H_8O^+$ | **71.04914** | 200 | 0.29 | $C_4H_7O^+$ |
| | 89.05971 | 90 | 0.03 | | 41.0383 | 58 | 0.08 | $C_3H_5^+$ |
| | 43.05423 | 83 | 0.03 | $C_3H_7^+$ | | | | |
| Propanal | **57.03349** | 7606 | 0.85 | $C_3H_5O^+$ | **57.03349** | 3388 | 0.74 | $C_3H_5O^+$ |
| | 88.0393 | 692 | 0.08 | $C_3H_6NO_2^+$ | 37.0284 | 782 | 0.17 | |
| | 75.04405 | 370 | 0.04 | $C_3H_7O_2^+$ | 59.04914 | 401 | 0.09 | $C_3H_7O^+$ |
| | 59.04914 | 288 | 0.03 | $C_3H_7O^+$ | | | | |

**Figure S1: Wind rose for the measurement period in the dry season (left) and the wet-to-dry transition (right) of 2019.**

**Figure S2: A map of the ATTO site adopted from a map created by Andrew Crozier.**

**Figure S3: Median averaged time series of meteorological parameters. Temperatures were measured inside the canopy at 26 m and PAR was measured from the top of the 80-m walk-up tower. The shadings indicate the quartiles (25th and 75th).**

**Figure S4: Precipitation before and during the measurement periods marked with purple dashed lines. Precipitation was measured on the 325-m tall tower.**

**Figure S5: Pearson correlation coefficients for the observed carbonyl compounds with black carbon measured at 325 m on the tall tower.**

**Figure S6: Time series of acetaldehyde mixing ratios in the wet-to-dry transition season and the dry season of 2019 measured at the three sampling heights with an applied E/N of 70 Td (2019-06-23 to 2019-07-03 and 2019-09-27 to 2019-10-06) and 120 Td (2019-07-03 to 2019-07-08 and 2019-10-06 to 2019-10-14). Measured PAR is given in light blue color with values on the right axis.**

**Figure S7: Time series of acetone mixing ratios in the wet-to-dry transition season and the dry season of 2019 measured at the three sampling heights with an applied E/N of 70 Td (2019-06-23 to 2019-07-03 and 2019-09-27 to 2019-10-06). Measured PAR is given in light blue color with values on the right axis.**

**Figure S8: Time series of propanal mixing ratios in the wet-to-dry transition season and the dry season of 2019 measured at the three sampling heights with an applied E/N of 70 Td (2019-06-23 to 2019-07-03 and 2019-09-27 to 2019-10-06) and 120 Td (2019-07-03 to 2019-07-08 and 2019-10-06 to 2019-10-14). Measured PAR is given in light blue color with values on the right axis.**

**Figure S9: Time series of methyl ethyl ketone mixing ratios in the wet-to-dry transition season and the dry season of 2019 measured at the three sampling heights with an applied E/N of 70 Td (2019-06-23 to 2019-07-03 and 2019-09-27 to 2019-10-06). Measured PAR is given in light blue color with values on the right axis.**

**Figure S10: Time series of methacrolein mixing ratios in the wet-to-dry transition season and the dry season of 2019 measured at the three sampling heights with an applied E/N of 70 Td (2019-06-23 to 2019-07-03 and 2019-09-27 to 2019-10-06) and 120 Td (2019-07-03 to 2019-07-08 and 2019-10-06 to 2019-10-14). Measured PAR is given in light blue color with values on the right axis.**

**Figure S11: Time series of methyl vinyl ketone mixing ratios in the wet-to-dry transition season and the dry season of 2019 measured at the three sampling heights with an applied E/N of 70 Td (2019-06-23 to 2019-07-03 and 2019-09-27 to 2019-10-06). Measured PAR is given in light blue color with values on the right axis.**

**Figure S12: Time series of C5-aldehyde mixing ratios in the wet-to-dry transition season and the dry season of 2019 measured at the three sampling heights with an applied E/N of 70 Td (2019-06-23 to 2019-07-03 and 2019-09-27 to 2019-10-06) and 120 Td (2019-07-03 to 2019-07-08 and 2019-10-06 to 2019-10-14). Measured PAR is given in light blue color with values on the right axis.**

**Figure S13: Time series of hexanal mixing ratios in the wet-to-dry transition season and the dry season of 2019 measured at the three sampling heights with an applied E/N of 70 Td (2019-06-23 to 2019-07-03 and 2019-09-27 to 2019-10-06). Measured PAR is given in light blue color with values on the right axis.**

**Figure S14: Time series of hexenal mixing ratios in the wet-to-dry transition season and the dry season of 2019 measured at the three sampling heights with an applied E/N of 70 Td (2019-06-23 to 2019-07-03 and 2019-09-27 to 2019-10-06) and 120 Td (2019-07-03 to 2019-07-08 and 2019-10-06 to 2019-10-14). Measured PAR is given in light blue color with values on the right axis.**

**Figure S15: Time series of benzaldehyde mixing ratios in the wet-to-dry transition season and the dry season of 2019 measured at the three sampling heights with an applied E/N of 70 Td (2019-06-23 to 2019-07-03 and 2019-09-27 to 2019-10-06) and 120 Td (2019-07-03 to 2019-07-08 and 2019-10-06 to 2019-10-14). Measured PAR is given in light blue color with values on the right axis.**

**Figure S16: Median averaged timeseries in the wet–to–dry transition season (June/July) of 2019 measured at all sampling heights for each hydrocarbon and its respective vertical profile at noon (12:00–15:00 LT) to the right. The shadings indicate the quartiles (25th and 75th). In the box-and-whisker plots, the boxes also represent the quartiles, while the residual data except for outliers are included in the whiskers. The detection limit (3 sigma) is indicated by dashed, black lines. The mixing ratios in gray font were calculated based on k-rate.**

**Figure S17: Median averaged timeseries in the dry season (September/October) of 2019 measured at all sampling heights for each hydrocarbon and its respective vertical profile at noon (12:00–15:00 LT) to the right. The shadings indicate the quartiles (25th and 75th). In the box-and-whisker plots, the boxes also represent the quartiles, while the residual data except for outliers are included in the whiskers. The detection limit (3 sigma) is indicated by dashed, black lines. The mixing ratios in gray font were calculated based on k-rate.**

---

## Author Response (AR1)

Dear Editor Frank Keutsch,

Thank you for the reviews provided. We are very grateful to the reviewer for the constructive, evaluation of the manuscript. All points raised by the reviewer have been addressed in the revised manuscript and the supplementary material, as detailed in the following text.

**Response to review 1**

In this work, Ringsdorf et al. describe atmospheric measurements of carbonyl containing OVOCs in the Amazon Rainforest using PTR-ToF-MS and NO⁺ CIMS. The use of PTR-ToF-MS and reagent switching enables differentiation between ketones and aldehydes which in turn enables the determination of their atmospheric fate, including reactivity and dry deposition. This includes vertically resolved measurements at 80-325 m of multiple species.

The paper exploits reagent ion switching to obtain information about ketones and aldehydes separately which is relatively novel.

The OVOC behavior described in this article is consistent with previous data, so although the measurements are somewhat novel there are no significant new conclusions.

The methods are described thoroughly, including calibrations or sensitivity estimates as well as possible interferences from isomers or decomposition of other species such as peroxides.

The authors do a very good job in contextualizing the measurements of each individual OVOC being reported. They cite the literature extensively and pose multiple hypotheses for observed diurnal and height variations in measurements.

Although reproducibility is impossible with field data, the authors do a great job detailing instrumental parameters such as E/N and sensitivity calculations which will enable future measurements to directly compare their results to the ones herein.

In terms of structure and presentation, the article has a good title and abstract which reflects the contents. The abstract and sections are logical and well organized, and the writing is great. Regarding content, most of the text is dedicated to previous work about the observed OVOCs making it seem more like a merge between a review and a measurement report. This level of background detail makes for a nice introductory read but it is not new science per se.

Regarding the supplementary data, I think the paper would benefit from some time series data. The only measurements presented in the article are averaged diurnal profiles and correlation tables. Presenting some time series data could provide further insight into the sampling height differences, reagent ion switching and differences between seasons which would strengthen the article.

As of now the article can be published with technical corrections.

The individual detection of carbonyl compound mixing ratios with a high temporal resolution in the Amazon presents a unique dataset, that advances the characterization of the BVOC diversity found in rainforest environments. By comparing this dataset to background literature the impression is created that all species were observed before, however, it is important to note that the instrumental methods did not allow the individual measurement of isomeric carbonyl compounds. To stress this point we adopted the following passages:

Line 467: *"Accordingly, in 2013, measurements of acetaldehyde using a PTR-quadrupole-MS (nominal m/z 45) vertical gradients below 80 m at ATTO showed increasing acetaldehyde between 24 m (inside the canopy, high influence by surrounding trees) and 79 m."*

Line 512: *"Both studies deployed PTR-quadrupole-MS operated with $H_3O^+$ reagent ions with a nominal mass resolution."*

We added time series of the carbonyl compounds discussed in the manuscript as suggested by reviewer 1 to the supplementary (Fig. S6-S15), to illustrate reagent ion switching and day-to-day variabilities.

Line 345: *"Time series of the aldehydes and ketones are provided in the supplementary (Fig. S6-S15)."*

During the review process, we noted that Fig. 4 included the Pearson correlation coefficients of the carbonyl species and ethanol for the dry season 2019, although ethanol mixing ratios were found below the detection limit (see Fig. S17). We changed the figure and its introduction in the manuscript accordingly:

Line 390: *"Figures 3-4 show the Pearson correlation coefficients (p) for both seasons divided into day (10:00–17:00 LT) and nighttime (22:00–05:00 LT) between the carbonyl compounds and between carbonyls and other selected VOC, including terpenes (isoprene, sum of monoterpenes), alkenes ($C_5$-alkenes, benzene), and oxygenated compounds (ethanol, furan, acetic acid, $C_5H_4O_3$), when measured above the detection limit."*

Minor comments:

Line 333: reference to chapter 4 should be removed.

Under the reasonable assumption of a carbonyl source at canopy level (based on emission inventories discussed in section 4)…

Figures S3-S5 are incorrectly ordered.

We thank the reviewer for noticing that detail, the numbers of figures S3 and S5 are changed accordingly.

> Figure S2 text boxes are illegible.

We decided to show an enlarged view of the ATTO site on the map without the extra information provided in the text boxes, as those are only interesting from a logistical point of view.

> Figure S4 could benefit should be replaced with a higher resolution version.

We agree, Fig. S4 is now replaced by a higher resolution figure.

> Figure S5 shows a plot of benzene with no data and non-sensical axes.

We thank the reviewer for pointing this out, we removed the concerning subplot ($C_5$-alkenes) as suggested.

**Response to review 2**

> Review of "Investigating Carbonyl Compounds above the Amazon Rainforest using PTR-ToF-MS with $NO^+$ Chemical Ionization" by Akima Ringsdorf et al. The manuscript analyzes the characteristics of different carbonyl compounds and emphasizes the necessity of supplementing the NO+ CIMS method to measure carbonyl compounds. The authors examine the sources and sinks in the Amazon rainforest of the selected carbonyl species in combination with a sufficient literature review. The article is substantial, with many citations and argumentation work. However, the writing focus of the discussion section of the manuscript is not clear enough, and the data analysis method is relatively simple. I recommend a major revision before its publication. My concerns in the manuscript are mainly listed below:

**Specific comments:**

1. **Line 99-100:** What is the temporal variation of humidity during the observation? Even with $NO^+$ ionizing chemistry, the reported $C_2$-$C_9$ carbonyl compounds are greatly affected by humidity. The author needs to specify whether the signal of these compounds has been calibrated for humidity.

Indeed, all calibrations and the fragmentation tests for carbonyl compounds were performed with moisturized synthetic air, which corresponds to a relative humidity of about 80 %, based on laboratory tests. This was chosen as it is in the same range of ambient humidity levels experienced at the ATTO site (see figure below, showing the diel cycle of RH during the time considered in the manuscript). The sensitivity of the carbonyl signals towards water originates from the formation of $H_3O^+$ ions (and ionized water clusters) that compete with $NO^+$ and from the formation of $NO^+$ water clusters. We accounted for the humidity-dependent formation of $(H_2O)NO^+$ by normalizing the signals to $NO^+$ and $(H_2O)NO^+$ (see line 176). The formation of $H_3O^+$ was kept low with impurities below 5 % during all measurements and laboratory tests.

[Figure]

To clarify this point, we have added the following sentence in line 160: *"To identify the distribution of product ions and fragments of carbonyls for the type of instrument used in this study, a single-compound headspace analysis was performed in the laboratory under humid conditions using a PTR-ToF-MS 8000. This is important as the sensitivity of the carbonyl signals towards water originates from the formation of $H_3O^+$ ions (and ionized water clusters) that compete with $NO^+$ and from the formation of $NO^+$ water clusters. It should be noted that we accounted for the humidity-dependent formation of $(H_2O)NO^+$ by normalizing the signals to $NO^+$ and $(H_2O)NO^+$."*

Line 188: *"The calibration was performed using moisturized synthetic air mixed with the VOC gas standard to mimic tropical conditions with 70 to 95 % relative humidity, typical of the ATTO site."*

2. **Line 101-121:** What is the delay time of measured compounds?Does the time resolution of 20 s for $NO^+$ PTR-ToF-MS be able to guarantee that the measured signal of speciated compounds was from the actual atmosphere, not the tube residue?

The experimentally derived delay time, meaning the time it takes for a concentration spike introduced at 320 m to be detected by the instrument at the base of the tower, was 90 seconds. The ambient air is drawn into the instrument with a flow of 10 l/min and constantly analyzed to produce mass spectra in the PTR-ToF-MS. A time resolution of 20 seconds only indicates the time over which those constantly recorded mass spectra are summed up and saved. We chose 20 seconds to achieve sufficient sensitivity and still be able to see sub-minute changes if needed. It is always possible to take an average over longer periods than the time resolution of the measurement, which we did by averaging to 4 minutes. This study does not report carbonyl variabilities within seconds or minutes, but rather daily variabilities. The effect of the 90-second delay time is a broadening of sharp concentration peaks with durations less than a few minutes, which are not expected and not the subject of this study. Tube residues are not measured; but any inlet effects from tube residues are minimized by a 5-minute flushing period before the actual sampling period starts. The reviewer is right, we did not report that the data was averaged to 4 minutes.

We now add this important information to line 119: *"VOC were measured by a Proton Transfer Reaction Time of Flight Mass Spectrometer (PTR-ToF-MS 4000, Ionicon Analytik, Innsbruck, Austria) (Jordan et al., 2009) with a time resolution of 20 seconds and averaged to 4 minutes."*

3. **Line 141-143:** The author needs to specify the parameter setting for NO$^+$ mode, including the ion source voltage and drift tube voltage since the difference of parameter setting will affect the abundance of impurity ions (H$_3$O$^+$, O$_2^+$, and NO$_2^+$). And the author also needs to give the abundance of impurity ions during the campaign, especially for the ratio of O$_2^+$/NO$^+$, due to O$_2^+$ will cause interference when using NO$^+$ ionization for measurement.

We thank the reviewer for pointing this out, as so far, we only reported the E/N values, which indicate the degree of fragmentation. We added the relevant parameters to the manuscript, including the drift tube voltage U$_{drift}$ and source voltage U$_{source}$. We now also report the impurities of O$_2^+$, H$_3$O$^+$ and NO$_2^+$, derived from m/z 31.9893, m/z 21.0221*500, and m/z 46.9906*222 respectively.

Line 143: "*Two settings with varying E/N (electrical field strength to gas number density) values were applied. One set had a relatively low E/N of 70 Td (Air (NO) = 9 sccm, U$_{drift}$ = 500 V, p$_{drift}$ = 3.4 mbar, T$_{drift}$ = 60 deg C, U$_{source}$ = 70 V), which has been recommended in previous studies to minimize fragmentation (Koss et al., 2016; Romano and Hanna, 2018); the other was operated with 120 Td (Air (NO) = 9 sccm, U$_{drift}$ = 850 V, p$_{drift}$ = 3.4 mbar, T$_{drift}$ = 60 deg C, U$_{source}$ = 70 V) for comparison. Low impurities of H$_3$O$^+$ ($\leq$ 1 %), O$_2^+$ (< 0,1 %), and NO$_2^+$ (< 2.5 %) were achieved using both settings.*"

4. **Line 187-197:** It seems that the parameterized quantitative method proposed by Cappellin et al. is based on the proton transfer reaction between organic compounds and H$_3$O$^+$ I wonder if this method is suitable for NO$^+$ chemistry. Because NO$^+$ ions can react with organic compounds with multiple ways and occur simultaneously. I think the author needs to reconsider this issue. In addition, it is suggested that the author use a formula to explain how to obtain the sensitivity for those compounds not included in the gas standard based on k-rate.

The method for quantification using the respective reaction rate of the analyte and the primary ion, as described by Cappellin et al. for H$_3$O$^+$, calculates the efficiency of the chemical ionization reaction under drift tube conditions. The method can be adopted for all chemical ionization reactions taking place under controlled conditions, and has also been used for other CIMS instruments (for example in Heinritzi et al., 2016). Usually, the reaction rate was experimentally determined for H$_3$O$^+$, NO$^+$ and O$_2^+$ in a reaction chamber or flow tube (e.g. Španěl et al., 1997). As the reviewer pointed out, it has to be carefully considered that multiple reactions and fragmentation occur for the different analytes. This is addressed by the headspace analyses of single species as described in lines 159-169 and 193-208. The contributions of the different product ions are presented in Table S1.

To clarify this important point, the paragraph in line 194-206 now reads as follows.

"*For those compounds not included in the gas standard, mixing ratios were obtained by calculating the ionization efficiency with a previously determined reaction rate of NO$^+$ and the target compound under the current conditions in the drift tube (k-rate analysis) (Cappellin et al., 2012).*"

$$[VOC] = \frac{1}{c\,k\,t}\frac{[VOC^+]_{ncps}}{[NO^+]_{ncps}} \qquad\qquad (1)$$

*Here, k is the reaction rate, and t represents the reaction time in the drift tube, which can be approximated using the length of the drift tube, the mobility of the primary ions and the applied drift voltage. Using Equation 1, the mixing ratio of a VOC is calculated from the normalized measured signal (ncps = normalized counts per second) of the main product ion. However, the reaction rates (k-rates), also presented in Table 1, have been experimentally derived for the sum of all product ions. Thus, a weighting factor c for the relative production of the target ion needs to be applied, which was also obtained by the single-compound headspace analysis from the slope of the signals of the target ion vs. other product ions. The mixing ratios of both E/N settings, obtained by applying Equation 1 with the respective product ion distributions, agree well for most compounds (except for n-hexanal and ketones, which have a low sensitivity at 120 Td)."*

We thank the reviewer for this point as introducing the c factor has helped make the explanation more coherent.

5. **Table 1 and Table S1:** According to previous studies for the reaction pathway of $NO^+$ to organic compounds (Charge transfer ($M^+$), hydride abstraction ($M^+$- H), association reaction ($MNO^+$) or hydroxide ion transfer ($M^+$- OH)) should occur simultaneously. However, the differences in ionization energy (IE) and the chemical bond can cause species to react more easily with $NO^+$ in one particularly pathway, associated with the formation of fragments. Therefore, we need to identify the characteristic product ions (which refers to the ion formula the author shown in the table) according to the contribution of each product ion after reaction. The author should show this contribution here, which also be better explain the influence of different E/N conditions to measurements.

If we have understood correctly, the reviewer asks for the inclusion of the product ions and their relative contribution for each carbonyl species resulting from the single headspace lab experiment (which is currently given in Table S1) in Table 1. We believe this would be too much information to summarize into one table, making it unwieldy. Nevertheless, to show the influence of the E/N on the product ions, as the reviewer requested, the weighting factor *c* is included in this table. *c* represents the contribution of the main product (or parent) ion to the sum of all product ions and was introduced in Equation 1.

6. **Line 268-285:** What is the contribution of local emissions and long-range transport during the measurement in dry season? As mentioned above that region is affected by long-range transport from African biomass burning pollution. And I prefer to see an overall picture of carbonyl emissions with concentrations and contributions.

We fully agree with the reviewer that given the relatively long atmospheric lifetimes of some of the carbonyl compounds (e.g. weeks) a remote biomass burning contribution to the signals from elsewhere in the hemisphere is possible. Indeed, long-range transport of aged biomass-burning plumes has been reported from ATTO site measurements (mainly from South America as shown in Holanda et al., 2023). However, observing diurnal cycles of all carbonyl compounds (see Figure

1 and 2) suggests the dominance of light and temperature-based biogenic emission and formation from the rainforest. We did not find a biomass burning tracer in the VOC data; acetonitrile is detected with a low sensitivity by NO$^+$ (Koss et al., 2016), with most values below the detection limit. Black carbon is simultaneously measured at ATTO and can indicate the pollution level from burning emissions, which was higher in the dry season, as expected. By correlating with BC, we found that some masses had correlations with Pearson coefficients > 0.55 (line 319), which suggests a possible contribution from fire plumes on top of the primary and secondary rainforest sources. To disentangle emissions, chemistry, and transport, we would need a known ratio of primarily emitted tracers that is also sensitive to fire plumes to quantify the contribution of fire emissions in the air mixture, or we would need to know the emission fluxes. These flux measurements are planned for the coming years. The current study focuses on the observed concentration profiles followed by a detailed discussion of their sources, including long-range transport of biomass burning plumes based on correlations (black carbon) and previous literature. We think that a further model analysis of global or intercontinental biomass burning transport is beyond the scope of this paper.

7. **Line 298-309:** Just a suggestion, maybe the author could further quantify the influence of biomass burning from Africa and South America based on PAN, which is basically present in aging plumes.[1]

Following the reviewer's suggestion, we looked for PAN ionized by NO$^+$ via association reaction, hydride ion abstraction, charge transfer, and proton transfer but found no signals. Nevertheless, we thank the reviewer for the idea. After careful consideration, we concluded that, unfortunately PAN is a rather unspecific tracer. In addition to forming in biomass burning plumes, it can be formed in cities like ozone as a secondary product of the photochemical processing of anthropogenic VOC and NOx. Additionally, where NOx and BVOC are co-present, PAN can be formed from the photochemical processing of the BVOC and NOx (see for example Williams et al., 1997). Moreover, whether PAN is detectable in a biomass burning plume will depend on the altitude the plume is transported. Being thermally unstable, PAN will decompose if transport occurs in the boundary layer where the measurements take place.

8. **Line 334:** Short-lived aldehydes do not include isoprene.

Agreed, to avoid confusion we now changed the sentence to: "…while levels of short-lived aldehydes will tend to be zero at higher altitudes, analogous to isoprene."

9. **Line 440-443:** The authors believe that ethanol was formed by anaerobic reactions on the surface, but why does the ethanol concentration decrease with increasing altitude during the transition season?

We are grateful for the reviewer's careful examination of the manuscript; this is an excellent point and something we have puzzled over. It is counterintuitive that ethanol shows the highest median mixing ratios at 325 m although the main contribution to the ethanol budget was found to be from plants (Kirstine and Galbally, 2012). Below we show the time series of ethanol in the transition season. On several days, we observed similar mixing ratios throughout the sampling heights (28.06, 02.07, 06.07.2019). On the other days, we saw and reported increasing mixing ratios with

height. Our analysis proceeded as follows: we excluded biomass burning as a major contribution to ethanol since the mixing ratios in the dry season were low and showed less variability with altitude. The vertical ethanol distribution in the transition season could be related to transport from the more flood-impacted riverside. At night, such emissions would remain in the residual layer (150 and 325 m) whereas ethanol is deposited to the canopy in the stable nocturnal boundary layer (80 m). The daily evolution shown in Figure S16 supports this hypothesis. The median mixing ratio of ethanol at night at 80 m decreased throughout the night, while the mixing ratios at 150 and 325 m were not. Of course, other yet unknown sources may exist as well. Ethanolic fermentation does not only occur in anerobic soils and roots but generally depends on the oxygen availability within tissues and organisms. Within this context the ethanolic fermentation with lichens during period of high Thallus water content may demonstrate our gaps in understanding (Wilske et al., 2001). The exact mechanisms determining the ethanol vertical profile remain unclear, and we agree that we cannot attribute the seasonal change of ethanol to ethanolic fermentation during root flooding only.

The text was thus amended:

Line 461:" *In this study, a strong correlation was found for ethanol and acetaldehyde in the nighttime during the transition season (p = 0.92). The high correlation coefficient at 80 m could originate from similar sinks, such as deposition to the canopy or related sources, such as the ethanolic fermentation pathway. Ethanol mixing ratios were ten times higher in the transition season and showed a diel maximum at nighttime. Since river levels were at their maximum levels in the transition season, root flooding may be partially responsible for the seasonal variability of ethanol (Kirstine and Galbally, 2012). However, acetaldehyde showed a different seasonal variability, indicating that other sources than those of ethanol were dominant.*"

[Figure]

10. **Line 498-545:** During the daytime, acetone and C5-ketones have stronger vertical gradients than the more reactive isoprene and monoterpenes. This phenomenon is very important for analyzing the sources and sinks of ketones. On a well-mixed daytime, it is difficult to understand the strong vertical gradient of ketones, even though the ketones have a strong source at the surface, as the authors believe.

Again, we are grateful for the reviewer's insightful comment. Indeed, acetone and $C_5$-ketones' vertical profiles show different vertical distributions throughout the three sampling heights than isoprene and monoterpenes. The measurements indicate that at 150 and 325 m the less reactive ketones tend to be well mixed as expected. However, the roughness sublayer (layer most influenced by the canopy) sampled at 80 m showed distinctly higher mixing ratios. In contrast, the more reactive isoprene and monoterpenes are decreasing above 150 m. As the reviewer correctly points out, the total observed relative decrease between 80 and 325 m is larger for acetone and $C_5$-ketones compared to isoprene and monoterpenes, due to their large gradient between 80 and 150 m. This is indeed puzzling, so we analyzed the data as follows: first, we went through possible measurement biases, which could have led to the observed large concentration gradients of ketones, to rule them out.

1. Contamination of the ketone signals at 80 m from short-lived substances (e.g sesquiterpenes)
2. Line loss of ketones

Contamination of the ketone signals means the detection of fragments of other molecules on the same mass as the respective ketone (see section 2.4). The fragmentation of other molecules takes place primarily in the drift tube. Especially at the low E/N of 70 Td, which is applied for detecting ketones, fragmentation is expected to be minimal. We found no hints of acetone contamination in reaction studies of different molecules with $NO^+$, and the acetone detected in Boulder, CO, showed no interference from other molecules (supplementary of Koss et al., 2016). We also compared the vertical distribution of acetone to measurements using $H_3O^+$ as a primary ion. This is possible since its isomer, butanal, was not present in the Amazon. The distribution agrees well with the acetone presented in this study, indicating no contamination of both product ions.

For $C_5$-ketones we found molecules that could potentially interfere with their signal, but none of them was reported to be abundant in tropical forests, especially in the first 100 m of the boundary layer.

The formation of acetone and $C_5$-ketones from ozonolysis in the inlet line was considered to be of minor importance as a possible bias as discussed in line 115. Other line effects to artificially produce ketones (e.g from the tubing surfaces) can also be ruled out as it would occur in all three inlet lines, and be even stronger in the longer lines.

The opposite, line loss of ketones is not feasible, since the $C^*$ of acetone and $C_5$- ketones are in the same range as the other carbonyl species (supplementary of Li et al., 2023). We did also not observe any loss of acetone when injecting the VOC gas standard into the 325 m inlet line.

After ruling out possible measurement biases, we can speculate on the possible mechanisms that result in the observed vertical distribution. In the case of isoprene and monoterpenes, the situation appears straightforward: emission occurs from the canopy, and concentrations decrease with height due to chemical oxidation and dilution caused by mixing from above. The picture for carbonyls is much more complicated. Direct emission can occur from the canopy but secondary formation can also take place, either through gas phase oxidation by OH or $O_3$ (e.g. through the oxidation of pinene), or via ozone reacting on leaf surfaces. Importantly, the canopy can also

uptake carbonyl compounds, as was shown by (Edtbauer et al., 2021; Kesselmeier, 2001; Rottenberger et al., 2004). The multiple sweeps and ejections of air in and out of the canopy in the roughness sublayer can therefore make the carbonyl uptake very efficient. Therefore, the roughness layer measurements at 80 m represent the net effect of several competing processes. One possibility is that secondary formation of the carbonyls competes with uptake to generate a maximum above the canopy somewhere between 35m-100m. The strong uptake would then inhibit upward mixing to 150 and 320 m and thus generate the sharp profile measured. This means that the 50 m region directly above the canopy will be an extremely interesting region for future studies.

In fact we have already planned to examine VOC at ATTO with a similar but lighter PTR-ToF-MS that can be integrated into an elevator fixed to the side of the tower. This will allow continuous observations as a function of altitude and particularly near the surface.

We also considered possible atmospheric sinks other than chemistry, which could be condensation to liquid droplets, but this is unlikely to happen to an extent that can explain the gradient observed between 80 and 150 m.

To reflect this discussion in the manuscript, we add the following text:

Line 520: *"The vertical distribution of acetone showed clearly enhanced mixing ratios at 80 m during daytime compared to well-mixed conditions at the higher sampling points. The gradient in the first 150 m above the canopy is strong despite the low reactivity of acetone, which raised the question of how acetone is distributed vertically in the rough surface layer."*

Line 539: *"Based on the information obtained in 2013 and the observations from this study, secondary production in the dry and transition season appears to peak above the canopy, adding up to varying contributions of direct emissions and uptake by vegetation. It is thus possible that strong secondary formation competes with uptake by vegetation to generate a local maximum in the rough surface layer, which is observed in this study by the enhanced mixing ratios observed at 80 m. Sweeps and ejections in and out of the canopy in the roughness sublayer could make the uptake of acetone by different vegetation species and soils very efficient. The strong gradient between 80 and 150 m likely reflects an acetone peak in the vertical."*

Line 697:" As *suspected for acetone, the vertical distribution of C5-ketones might have been peaking around 80 m as a result of the bidirectional exchange in the canopy and secondary formation."*

Line 818:" *Interestingly, elevated ketone mixing ratios in the roughness sublayer observed at 80 m by day suggest a large source above or at canopy level, balanced with a surface uptake process. To examine these strong vertical gradients observed for some ketones, continuous measurements with altitude are planned using a PTR-ToF-MS installed on an elevator attached to the tower. This system will allow investigation of the exchange of VOC between canopy and atmosphere and reveal whether mixing ratios of acetone, MEK and $C_5$-ketones are peaking around 80 m as suggested by the observed elevated mixing ratios at 80 m."*

**Technical Corrections:**

1. **Table 1**: The references involved in the table may be annotated separately.

To improve the readability of table 1, we now added space between the k-rate and its reference. Adding another column and reporting the reference there would consume too much space and require a smaller font.

2. The picture can be named (for example, Fig. 1(a)), which can avoid the description of upper, lower, etc.

We thank the reviewer for this suggestion, but titles already name the subplots of all figures, so we do not reference them by upper, lower, etc.

3. **S6 and S7:** The naming format used for each species in the figures should be unified, now there are both species names, molecular formulas and ionic formulas

We now changed all titles to include the molecular formula and the species name.

4. **S7:** Why is the first picture empty?

The first subplot is empty as $C_5$-alkenes were not detected in the dry season of 2019. We now completely removed the subplot to not confuse the reader.

**Reference**

1. Liang, Y.; Weber, R. J.; Misztal, P. K.; Jen, C. N.; Goldstein, A. H., Aging of Volatile Organic Compounds in October 2017 Northern California Wildfire Plumes. *Environ Sci Technol* **2022,** *56*, (3), 1557-1567.

Cappellin, L., Karl, T., Probst, M., Ismailova, O., Winkler, P. M., Soukoulis, C., Aprea, E., Märk, T. D., Gasperi, F., and Biasioli, F.: On Quantitative Determination of Volatile Organic Compound Concentrations Using Proton Transfer Reaction Time-of-Flight Mass Spectrometry, Environ. Sci. Technol., 46, 2283–2290, https://doi.org/10.1021/es203985t, 2012.

Edtbauer, A., Pfannerstill, E. Y., Pires Florentino, A. P., Barbosa, C. G. G., Rodriguez-Caballero, E., Zannoni, N., Alves, R. P., Wolff, S., Tsokankunku, A., Aptroot, A., de Oliveira Sá, M., de Araújo, A. C., Sörgel, M., de Oliveira, S. M., Weber, B., and Williams, J.: Cryptogamic organisms are a substantial source and sink for volatile organic compounds in the Amazon region, Commun. Earth Environ., 2, 1–14, https://doi.org/10.1038/s43247-021-00328-y, 2021.

Heinritzi, M., Simon, M., Steiner, G., Wagner, A. C., Kürten, A., Hansel, A., and Curtius, J.: Characterization of the mass-dependent transmission efficiency of a CIMS, Atmospheric Meas. Tech., 9, 1449–1460, https://doi.org/10.5194/amt-9-1449-2016, 2016.

Holanda, B. A., Franco, M. A., Walter, D., Artaxo, P., Carbone, S., Cheng, Y., Chowdhury, S., Ditas, F., Gysel-Beer, M., Klimach, T., Kremper, L. A., Krüger, O. O., Lavric, J. V., Lelieveld, J., Ma, C., Machado, L. A. T., Modini, R. L., Morais, F. G., Pozzer, A., Saturno, J., Su, H., Wendisch, M., Wolff, S., Pöhlker, M. L., Andreae, M. O., Pöschl, U., and Pöhlker, C.: African biomass burning affects aerosol cycling over the Amazon, Commun. Earth Environ., 4, 1–15, https://doi.org/10.1038/s43247-023-00795-5, 2023.

Jordan, A., Haidacher, S., Hanel, G., Hartungen, E., Märk, L., Seehauser, H., Schottkowsky, R., Sulzer, P., and Märk, T. D.: A high resolution and high sensitivity proton-transfer-reaction time-of-flight mass spectrometer (PTR-TOF-MS), Int. J. Mass Spectrom., 286, 122–128, https://doi.org/10.1016/j.ijms.2009.07.005, 2009.

Kesselmeier, J.: Exchange of Short-Chain Oxygenated Volatile Organic Compounds (VOCs) between Plants and the Atmosphere: A Compilation of Field and Laboratory Studies, J. Atmospheric Chem., 39, 219–233, https://doi.org/10.1023/A:1010632302076, 2001.

Kirstine, W. V. and Galbally, I. E.: The global atmospheric budget of ethanol revisited, Atmospheric Chem. Phys., 12, 545–555, https://doi.org/10.5194/acp-12-545-2012, 2012.

Koss, A. R., Warneke, C., Yuan, B., Coggon, M. M., Veres, P. R., and de Gouw, J. A.: Evaluation of NO$^+$ reagent ion chemistry for online measurements of atmospheric volatile organic compounds, Atmospheric Meas. Tech., 9, 2909–2925, https://doi.org/10.5194/amt-9-2909-2016, 2016.

Li, X.-B., Zhang, C., Liu, A., Yuan, B., Yang, H., Liu, C., Wang, S., Huangfu, Y., Qi, J., Liu, Z., He, X., Song, X., Chen, Y., Peng, Y., Zhang, X., Zheng, E., Yang, L., Yang, Q., Qin, G., Zhou, J., and Shao, M.: Assessment of long tubing in measuring atmospheric trace gases: applications on tall towers, Environ. Sci. Atmospheres, 3, 506–520, https://doi.org/10.1039/D2EA00110A, 2023.

Romano, A. and Hanna, G. B.: Identification and quantification of VOCs by proton transfer reaction time of flight mass spectrometry: An experimental workflow for the optimization of specificity, sensitivity, and accuracy, J. Mass Spectrom., 53, 287–295, https://doi.org/10.1002/jms.4063, 2018.

Rottenberger, S., Kuhn, U., Wolf, A., Schebeske, G., Oliva, S. T., Tavares, T. M., and Kesselmeier, J.: Exchange of Short-Chain Aldehydes Between Amazonian Vegetation and the Atmosphere, Ecol. Appl., 14, 247–262, https://doi.org/10.1890/01-6027, 2004.

Španěl, P., Ji, Y., and Smith, D.: SIFT studies of the reactions of H3O+, NO+ and O2+ with a series of aldehydes and ketones, Int. J. Mass Spectrom. Ion Process., 165–166, 25–37, https://doi.org/10.1016/S0168-1176(97)00166-3, 1997.

Williams, J., Roberts, J. M., Fehsenfeld, F. C., Bertman, S. B., Buhr, M. P., Goldan, P. D., Hübler, G., Kuster, W. C., Ryerson, T. B., Trainer, M., and Young, V.: Regional ozone from biogenic hydrocarbons deduced from airborne measurements of PAN, PPN, and MPAN, https://doi.org/10.1029/97GL00548, 1997.

Wilske, B., Holzinger, R., and Kesselmeier, J.: Evidence for Ethanolic Fermentation in Lichens during Periods of High Thallus Water Content, 2001.